# FRQ-CK1 interaction determines the period of circadian rhythms in *Neurospora*

Xiao Liu[1,2,10], Ahai Chen[1,3,10], Angélica Caicedo-Casso[4], Guofei Cui[5], Mingjian Du[1], Qun He[5], Sookkyung Lim[6], Hang J. Kim[7], Christian I. Hong[8,9] & Yi Liu[1]

Circadian clock mechanisms have been extensively investigated but the main rate-limiting step that determines circadian period remains unclear. Formation of a stable complex between clock proteins and CK1 is a conserved feature in eukaryotic circadian mechanisms. Here we show that the FRQ-CK1 interaction, but not FRQ stability, correlates with circadian period in *Neurospora* circadian clock mutants. Mutations that specifically affect the FRQ-CK1 interaction lead to severe alterations in circadian period. The FRQ-CK1 interaction has two roles in the circadian negative feedback loop. First, it determines the FRQ phosphorylation profile, which regulates FRQ stability and also feeds back to either promote or reduce the interaction itself. Second, it determines the efficiency of circadian negative feedback process by mediating FRQ-dependent WC phosphorylation. Our conclusions are further supported by mathematical modeling and in silico experiments. Together, these results suggest that the FRQ-CK1 interaction is a major rate-limiting step in circadian period determination.

[1] Department of Physiology, University of Texas Southwestern Medical Center, Dallas, TX 75390-9040, USA. [2] State Key Laboratory of Mycology, Institute of Microbiology, Chinese Academy of Sciences, Beijing 100101, China. [3] Key Laboratory of Biopesticides and Chemical Biology, Ministry of Education, Fujian Agriculture and Forestry University, Fuzhou, Fujian 350002, China. [4] Department of Mathematics, Universidad del Valle, Cali 760032, Colombia. [5] State Key Laboratory of Agrobiotechnology and MOA Key Laboratory of Soil Microbiology, College of Biological Sciences, China Agricultural University, Beijing 100193, China. [6] Department of Mathematical Sciences, University of Cincinnati, Cincinnati 45221, USA. [7] Division of Statistics and Data Science, University of Cincinnati, Cincinnati 45221, USA. [8] Department of Pharmacology and Systems Physiology, University of Cincinnati, Cincinnati, OH 45267, USA. [9] Division of Developmental Biology, Cincinnati Children's Hospital Medical Center, 3333Burnet Avenue, Cincinnati, USA. [10]These authors contributed equally: Xiao Liu, Ahai Chen. Correspondence and requests for materials should be addressed to Y.L. (email: Yi.Liu@UTSouthwestern.edu)

Circadian clocks control daily rhythms of molecular and physiological activities to allow organisms to adapt to daily environmental changes. In *Neurospora*, *Drosophila*, and mammalian cells, the circadian oscillators consist of auto-regulatory transcription- and translation-based negative feedback loops[1–5]. In these negative feedback loops, Per-Arnt-Sim (PAS)-domain transcription factors act as positive elements that activate the transcription of the negative elements, and the latter feedback to repress their own transcription by inhibiting the activity of the positive elements. How circadian clocks generate endogenous and temperature compensated periodicity of ~24 h has always been a central question in chronobiological research. Although the mechanisms of eukaryotic circadian clocks have been extensively investigated in the past four decades, the main rate-limiting steps that determine circadian period length remains unclear[6,7].

Post-translational modification of clock proteins by phosphorylation is a highly conserved feature in eukaryotes. A major function of phosphorylation is to regulate the stability of the negative elements. Mutations of clock protein phosphorylation sites or of kinases alter protein stability and result in changes in period length[8–17], which led to the proposal that degradation rates of the negative elements is the major factor that determines the period length. However, protein stability of the negative elements frequently does not correlate with circadian period length in some mutants[18,19]. In addition, impairments in the ubiquitin proteasome pathway that is required for PERIOD (PER) degradation only result in modest period changes in animal cells[20–22]. Recently, it was shown that in *Neurospora* mutants with a defective FREQUENCY (FRQ) degradation pathway, FRQ degradation rate and circadian period length can be uncoupled[7,23]. This result suggested that FRQ phosphorylation but not protein stability plays an important role in period determination. How FRQ phosphorylation regulates circadian period length is unclear.

Multiple conserved kinases, including casein kinase 1 (CK1), CK2, and protein kinase A (PKA), phosphorylate FRQ in *Neurospora* and PER in animal systems. Among these kinases, CK1 has been shown to play a major role in FRQ and PER phosphorylation. Unlike typical kinase-substrate interactions, which are weak and transient, CK1 forms a tight stoichiometric complex with FRQ in *Neurospora* and with PER in *Drosophila* and mammals[12,14,24–29], suggesting that this conserved feature is critical for eukaryotic clock functions.

In the core *Neurospora* circadian negative feedback loop, the PAS domain-containing factors WHITE COLLAR (WC) 1 and 2 form a heterodimeric complex that activates *frq* transcription by binding to the C-box of its promoter[13,30–32]. FRQ forms a complex with FRQ-interacting RNA helicase (FRH) to act as the negative element[25,33]. After its synthesis, FRQ becomes progressively phosphorylated by CK1a (the *Neurospora* homolog of the mammalian CK1δ/ε), CK2, PKA, and other kinases at about 100 sites[13–15,34–37]. CK1a is a major FRQ kinase in vivo, but, unlike CK2 and other FRQ kinases, CK1a forms a tight complex with FRQ through two FRQ-CK1a interaction domains (FCD). Mutations of either FCD can completely eliminate the FRQ-CK1a interaction and abolish CK1a-mediated FRQ phosphorylation in vivo and in vitro[13,38,39]. Extensive FRQ phosphorylation triggers its degradation mediated by F-box/WD-40 repeat-containing protein-1 (FWD-1, the *Neurospora* homolog of Slimb and β-TRCP) that interacts with phosphorylated FRQ[40,41]. FRQ phosphorylation has opposing roles in period determination: Mutations of FRQ phosphorylation sites can result in short or long period mutants, and, in many cases, the period length correlates with FRQ stability[9,14,15]. However, it was recently shown that in the *fwd-1* mutant background, the FRQ degradation rates and its circadian period length are uncoupled, challenging the

long proposed role of FRQ degradation in circadian clock function[23].

To close the negative feedback loop, the FRQ-FRH complex interacts with the WC complex and recruits CK1a to phosphorylate WC proteins, a process that inhibits WC DNA binding[13,42–47]. When the FRQ-CK1a interaction is abolished, WCs remain hypophosphorylated and active. Therefore, the physical interaction between FRQ and CK1a has two functions in the clock: this interaction mediates FRQ phosphorylation, and it is required for the circadian negative feedback process. In this study, by using a combination of genetic, biochemical, mathematical modeling and in silico experiments, our results support that the formation of the FRQ-CK1a complex is the major rate-limiting process in circadian period determination.

## Results

**Period is correlated with FRQ-CK1a interaction but not FRQ stability.** To examine the correlation between FRQ stability and period length, we compared FRQ stability in previously described FRQ phosphorylation site mutants[14,15]. Two mutants at FRQ phosphorylation sites either downstream of the FCD-2 domain (M9) or within the PEST-1 domain (M10) both exhibited very long period phenotypes (29.8 and 32.1 h, respectively) (Fig. 1a). Comparison of FRQ stability after the addition of cycloheximide (CHX) showed that although the long period correlated with increased FRQ stability in the M9 mutant, similar FRQ stability was observed in the M10 mutant and in the wild-type strain (Fig. 1b), indicating that FRQ stability alone does not explain these long period phenotypes.

To determine the mechanism of period phenotypes in these mutants, we first compared the interaction between FRQ and WC-2, a critical step in the circadian negative feedback process. Immunoprecipitation of WC-2 showed that the amount of FRQ precipitated with WC-2 was similar after normalized with the Input FRQ level in the mutants and in the wild-type strain (Fig. 1c). Because CK1a forms a stable complex with FRQ and it not only phosphorylates FRQ but is also recruited by FRQ to mediate WC phosphorylation to close the circadian negative feedback loop[13,44,48], we examined the FRQ-CK1a interaction in these strains. Because FRQ phosphorylation is rhythmic in constant darkness (DD), we performed immunoprecipitation experiments using cultures grown in constant light (LL) which has constant FRQ phosphorylation profiles. Using a CK1a antibody that specifically recognizes the endogenous CK1a protein (Supplementary Fig. 1a), we demonstrated that the amount of FRQ associated with CK1a (normalized with the Input FRQ) was less in the mutants than the wild-type strain even though total FRQ levels in the mutants were higher (Fig. 1d). This indicated that the FRQ-CK1a interaction was significantly reduced in both mutant strains compared to the wild-type strain. In addition, the FRQ-CK1a interaction was lower in the M9 mutant (period 32.1 h) than in the M10 mutant (period 29.8 h). Thus, the reduced FRQ-CK1a interaction but not FRQ stability can potentially explain the long period phenotypes of both mutants. These results also suggest that the FRQ phosphorylation events impaired in these two mutants promote the FRQ-CK1a interaction.

We next examined two classic *frq* period mutants: *frq¹* and *frq⁷*[49]. *frq¹* is a short period mutant and *frq⁷* is a long period mutant, each due to a missense mutation in the FRQ open reading frame (Supplementary Fig. 1b). Their period phenotype was previously attributed to differences in FRQ stability: FRQ is more stable in *frq⁷* and is less stable in *frq¹*[50]. To evaluate the FRQ-CK1a interaction, we introduced a Myc-tagged CK1a expression construct into the mutants and wild-type strain. By

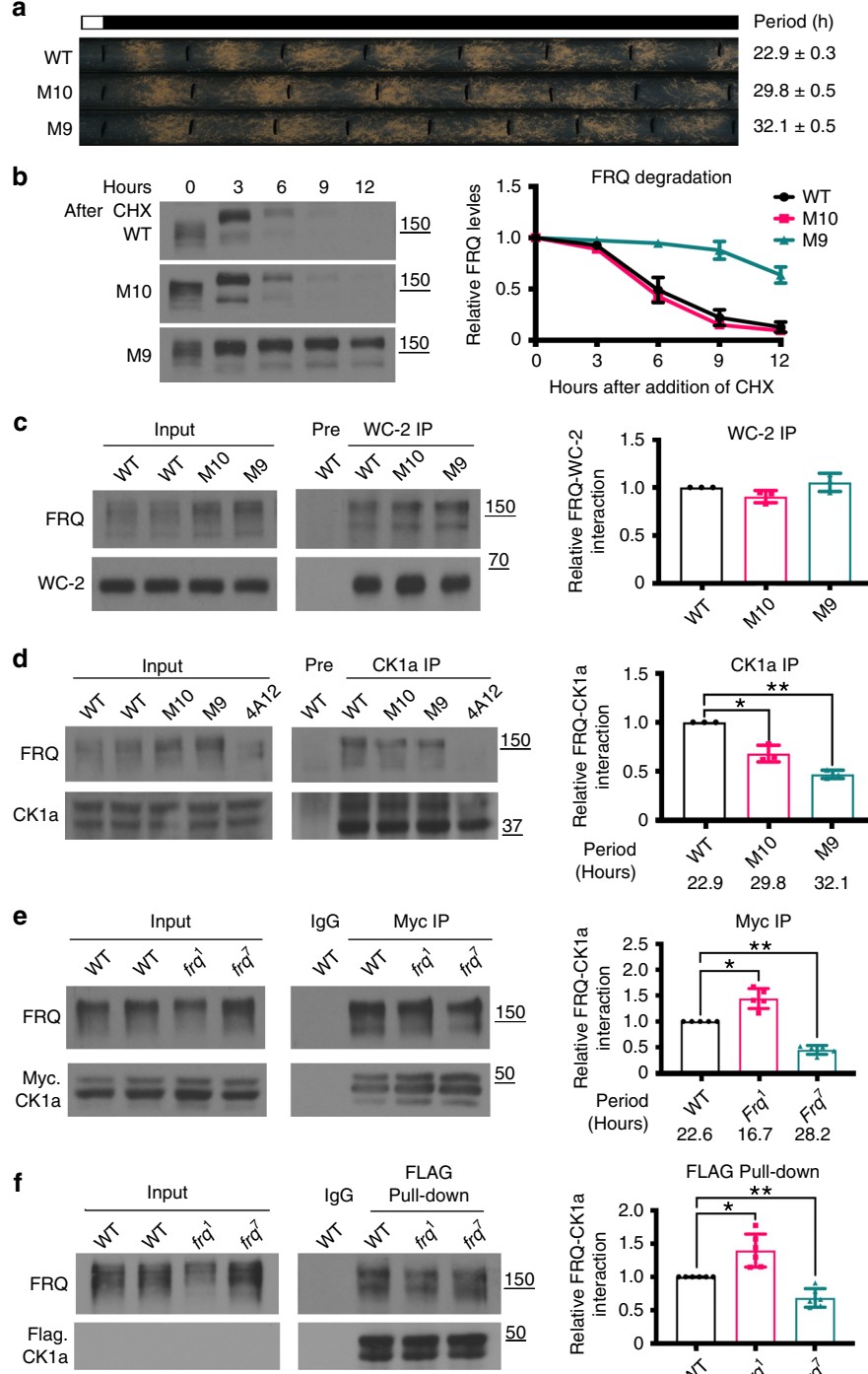

**Fig. 1** Period correlates with FRQ-CK1a interaction, but not with FRQ stability. **a** Race tube assay showing the circadian conidiation rhythms of two FRQ phosphorylation site mutants M9 and M10. Error bars are standard error of mean ($n = 6$). **b** Western blot analysis showing FRQ degradation profiles in the indicated strains after the addition of CHX. Both of hyper- and hypo-phosphorylated forms of FRQ are included in the quantification. The densitometric analysis results are shown on the right. Error bars indicate standard deviations ($n = 3$). The position of the molecular weight marker is indicated. **c** WC-2 immunoprecipitation assay in the M9 and M10 mutants. Quantification of relative FRQ-WC-2 interaction levels is based on the ratio of IP to Input and normalized with WC-2 level. The densitometric analysis is shown on the right. Error bars indicate standard deviations ($n = 3$). **d** CK1a immunoprecipitation assay in M9 and M10 mutants. Quantification of relative FRQ-CK1a interaction levels is based on the ratio of IP to Input and normalized with CK1a level. The densitometric analysis is shown on the right. Error bars are standard deviations ($n = 3$). *$P < 0.05$, **$P < 0.01$. Student's t-test was used. **e** c-Myc immunoprecipitation assay in the wild-type, $frq^1$, and $frq^7$ strains that express Myc.CK1a. Strains were cultured in LL. The densitometric is shown on the right. Error bars are standard deviations ($n = 5$). *$P < 0.05$, **$P < 0.01$. Student's $t$-test was used. **f** FLAG-CK1a pull-down in wild-type, $frq^1$, and $frq^7$ extracts. Recombinant FLAG-CK1a was incubated with the extracts, and anti-FLAG was used to immunoprecipitate FLAG-CK1a. Quantification of relative FRQ-CK1a interaction levels is based on the ratio of IP to Input. The densitometric analysis is shown on the right. Error bars are standard deviations ($n = 6$). *$P < 0.05$, **$P < 0.01$. Student's $t$-test was used. All strains were cultured in LL. Source data are provided as a Source Data file

analysis of total cell extracts prepared from cultures grown in LL, we found that levels of FRQ were reduced in the $frq^1$ strain and elevated in the $frq^7$ strain compared to the wild-type strain (Fig. 1d, e and Supplementary Fig. 1c). In addition, FRQ-WC interaction was not affected in these mutants (Supplementary Fig. 1d). Immunoprecipitation of Myc-CK1a showed that compared to the wild-type strain, the FRQ-CK1a interaction was significantly increased in the $frq^1$ mutant but was decreased in the $frq^7$ mutant (Fig. 1e).

To confirm this conclusion, the recombinant FLAG-tagged CK1a was incubated with *Neurospora* extracts from either the wild-type or mutant strains. Precipitation with the FLAG antibody showed that there was more FRQ associated with FLAG-CK1a in the $frq^1$ extract than in the $frq^7$ extract (Fig. 1f). These results indicate that the FRQ-CK1a interaction negatively correlates with period length in these classic clock mutants. Due to the role of FRQ-CK1a interaction in mediating FRQ phosphorylation and its subsequent degradation, the altered FRQ-CK1a interaction explains the different FRQ stability in $frq^1$ and $frq^7$ mutants. These results suggest that the altered FRQ-CK1a interaction, but not FRQ stability, is the primary cause of period phenotypes in these clock mutants.

**Mutating the FRQ-CK1a interaction domain drastically affects period**. Two FRQ amphipathic α helix-containing domains, FCD1 (aa319-326) and FCD2 (488–495), were previously shown to mediate the FRQ-CK1a complex formation (Fig. 2a). Mutations of either domain can completely abolish the FRQ-CK1a interaction, resulting in FRQ hypophosphorylation, increased FRQ stability, and arrhythmicity[13,38]. To directly demonstrate the role of the FRQ-CK1a interaction on circadian period length, we introduced single amino acid mutations into each of these two FCD α helixes and determined circadian periods of the mutant strains by race tube assays. Remarkably, circadian period lengths of these FCD mutants ranged from 24.7 to 48.0 h; some of the mutations, such as 4A11 and 4A12, resulted in arrhythmicity due to complete loss of the FRQ-CK1a interaction (Fig. 2b, c). These are some of the most severe circadian period effects ever reported for *Neurospora*. To our knowledge, the 48 h L488V mutant is the longest circadian period mutant reported for a eukaryotic organism. Despite their dramatic period differences, these mutants all exhibited robust and precise conidiation rhythms (Fig. 2b, c), suggesting that the main effect of the FRQ-CK1a interaction is on circadian period length.

Immunoprecipitation of CK1a from extracts of mutants showed that the period length negatively correlated with the FRQ-CK1a interaction: The longer the period length of the mutant, the weaker the interaction between FRQ and CK1a (Fig. 2d). In contrast, the FRQ-WC-2 interaction was not affected in these mutants (Supplementary Fig. 2a). These results suggest that the FRQ-CK1a interaction is a major determinant of circadian period length in *Neurospora*.

Comparison of FRQ phosphorylation profiles for cultures grown in constant light (LL) showed that FRQ became hypophosphorylated in strains carrying mutations in the FRQ FCDs (Fig. 2e). In addition, the degree of impact on FRQ phosphorylation correlated with the FRQ-CK1a interaction: The longer the period length, the more hypophosphorylated FRQ became. The 4A12 mutant was the most hypophosphorylated, and no FRQ-CK1a complex was detected in this mutant. FRQ in the L488V mutant (48 h period) was more hypophosphorylated than those in other mutants, consistent with its dramatic decrease of FRQ-CK1a interaction. Thus, unlike typical kinase-substrate reactions, CK1a must first bind tightly to FRQ prior to its phosphorylation of FRQ.

We compared the FRQ rhythmic phosphorylation profile between the WT and one of the FCD mutants (Q494N) in DD (Supplementary Fig. 2b). Consistent with the long period phenotype of the Q494N strain, the phase of its FRQ phosphorylation rhythm was markedly delayed compared to the WT strain. In addition, we found that period length correlated with FRQ stability in the above mutant strains (Fig. 2f). Furthermore, we also inducibly over-expressed CK1a in the WT and FCD mutants (Q494N and V320I) to compare their period length changes due to an increase of CK1a level. Although the induction of CK1a resulted in period shortening in all strains, the FCD mutants were more sensitive to CK1a over expression than WT (Supplementary Fig. 2c). These results suggest that the binding equilibrium of the FRQ-CK1a complex formation is important for period determination. Together, these results indicate that the FRQ-CK1a interaction is a critical step that determine the efficiency of FRQ phosphorylation, its stability and the period length.

**The dynamic roles of FRQ phosphorylation on FRQ-CK1a interaction**. Depending on the site, mutations of FRQ phosphorylation sites lengthen or shorten the circadian period length[9,14,15]. The importance of the FRQ-CK1a interaction on period determination suggested that FRQ phosphorylation can also regulate FRQ-CK1 interaction. Supporting this conclusion, the FRQ-CK1a interaction was impaired in both M9 and M10 mutants (Fig. 1d), indicating that FRQ phosphorylation at these sites promotes the interaction between FRQ and CK1a, resulting in their long period phenotypes.

CHX treatment induces FRQ hyperphosphorylation and degradation[9]. Therefore, we evaluated the FRQ-CK1a interaction in the wild-type strain after CHX treatment. Immunoprecipitation of Myc-CK1a showed that the relative FRQ-CK1a interaction was markedly reduced when FRQ was hyperphosphorylated (Fig. 3a). This result is consistent with previous studies[39,51], indicating that FRQ hyperphosphorylation inhibits the FRQ-CK1a interaction.

FRQ phosphorylation profiles exhibit a robust circadian rhythm: Newly synthesized FRQ is hypophosphorylated during early subjective morning, is progressively more phosphorylated during the day, and becomes hyperphosphorylated in the middle of subjective night[52]. Thus we compared the relative amount of FRQ associated with CK1a at different time points in constant darkness (DD). Consistent with a previously reported Mass spectrometry data[14], our IP experiments showed that the relative amount of FRQ associated with CK1a exhibited a circadian rhythm (Fig. 3b): It was lowest when FRQ was hyperphosphorylated (DD28) and highest when FRQ phosphorylation was intermediate (DD22). When FRQ was hypophosphorylated (DD16 and DD34), the amount was intermediate. This result suggests that the effect of FRQ phosphorylation can have opposing effects on complex formation: progressive FRQ phosphorylation during the early stage promotes the interaction between FRQ and CK1a, but extensive phosphorylation inhibits the interaction. In contrast, the FRQ-WC-2 interaction was not significantly affected by FRQ phosphorylation levels (Supplementary Fig. 3a).

S513, S519, S683, S685, S885, and S887 are some of the previously identified FRQ phosphorylation sites[9,14,15]. The strain with the mutation S513R and the previously described M13 strain (mutations at S683 and S685) resulted in long period phenotypes (Supplementary Fig. 3b). In contrast, the strain with the mutation S519A and the previously described M17 strain (mutations at S885 and S887) led to period shortening (Supplementary Fig. 3b). Compared to the wild-type strain, the relative level of the

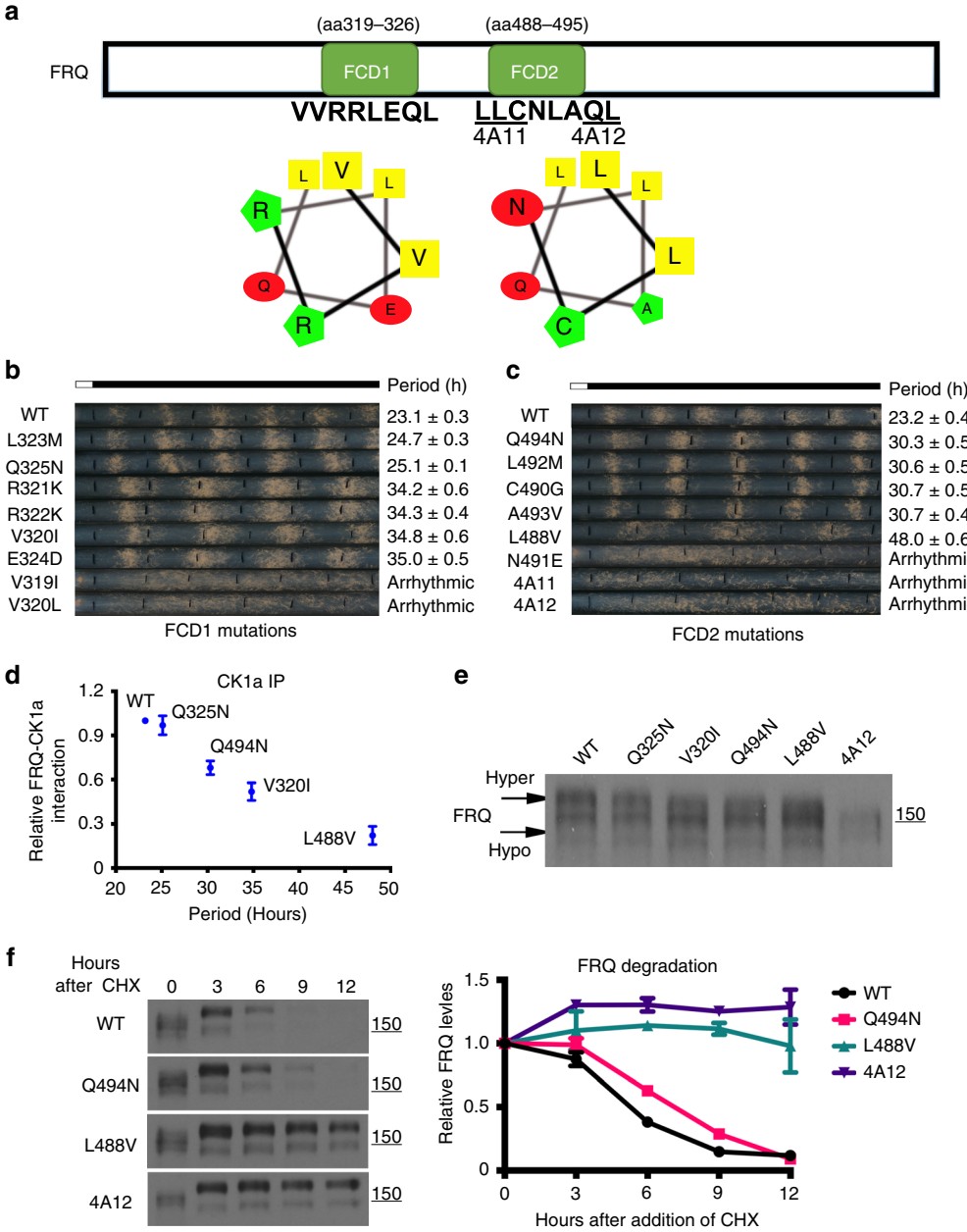

**Fig. 2** Dramatic impact on period by mutating the FCD domains. **a** A schematic diagram showing the FRQ FCD1 and FCD2 domains. The amino acid sequences within the domains that are critical for FRQ-CK1a interaction and helical representations are shown. The 4A11 and 4A12 mutations previously shown to completely disrupt the FRQ-CK1a interaction are indicated[45]. **b, c** Race tube assays showing the circadian conidiation rhythms and the period lengths of strains with mutation in **b** the FCD1 and **c** the FCD2 domains. Errors are standard errors of means ($n = 5$). **d** c-Myc immunoprecipitation assay showing that the amount of FRQ-Myc.CK1a in selected FCD period mutant strains relative to the amount in the wild-type strain. The strains were cultured in LL. Error bars are standard deviations ($n = 4$). **e** Western blot analysis comparing the FRQ phosphorylation profiles in the indicated strains. Strains were cultured in LL. Arrows indicates hyper- and hypo-phosphorylated FRQ. The position of the molecular weight marker is indicated. **f** Western blot analysis of FRQ in indicated strains after CHX treatment. Both hyper- and hypo-phosphorylated forms of FRQ are included in the quantification. The densitometric analysis is shown on the right. Error bars are standard deviations ($n = 3$). Source data are provided as a Source Data file

FRQ-CK1a complex was reduced in the S513R and M13 mutants but was increased in the S519A and M17 mutants (Fig. 3c, d), consistent with their period phenotypes. Moreover, the FRQ-WC-2 interaction was not affected in these mutants (Supplementary Fig. 3c–d).

FRQ is phosphorylated by CK2, but CK2 does not form a stable complex with FRQ[35,36,53]. The disruption of the *ckb* gene, which encodes the regulatory subunit of CK2, led to hypophosphorylation of FRQ and a lengthening of the period by about 6 h (Supplementary Fig. 3b)[35]. As expected, the FRQ-CK1a

interaction was also significantly reduced in the *ckb*$^{RIP}$ mutant, explaining its long period phenotype (Fig. 3e). Thus, FRQ phosphorylation at different sites can either positively or negatively regulate the FRQ-CK1a interaction. In addition, the period phenotypes of the FRQ phosphorylation site mutants correlate with the FRQ-CK1a interaction in different clock mutants.

We hypothesized that phosphorylation influences the conformation of FRQ, which in turn affects the binding of FRQ with CK1a. To test this, we compared limited trypsin digestion

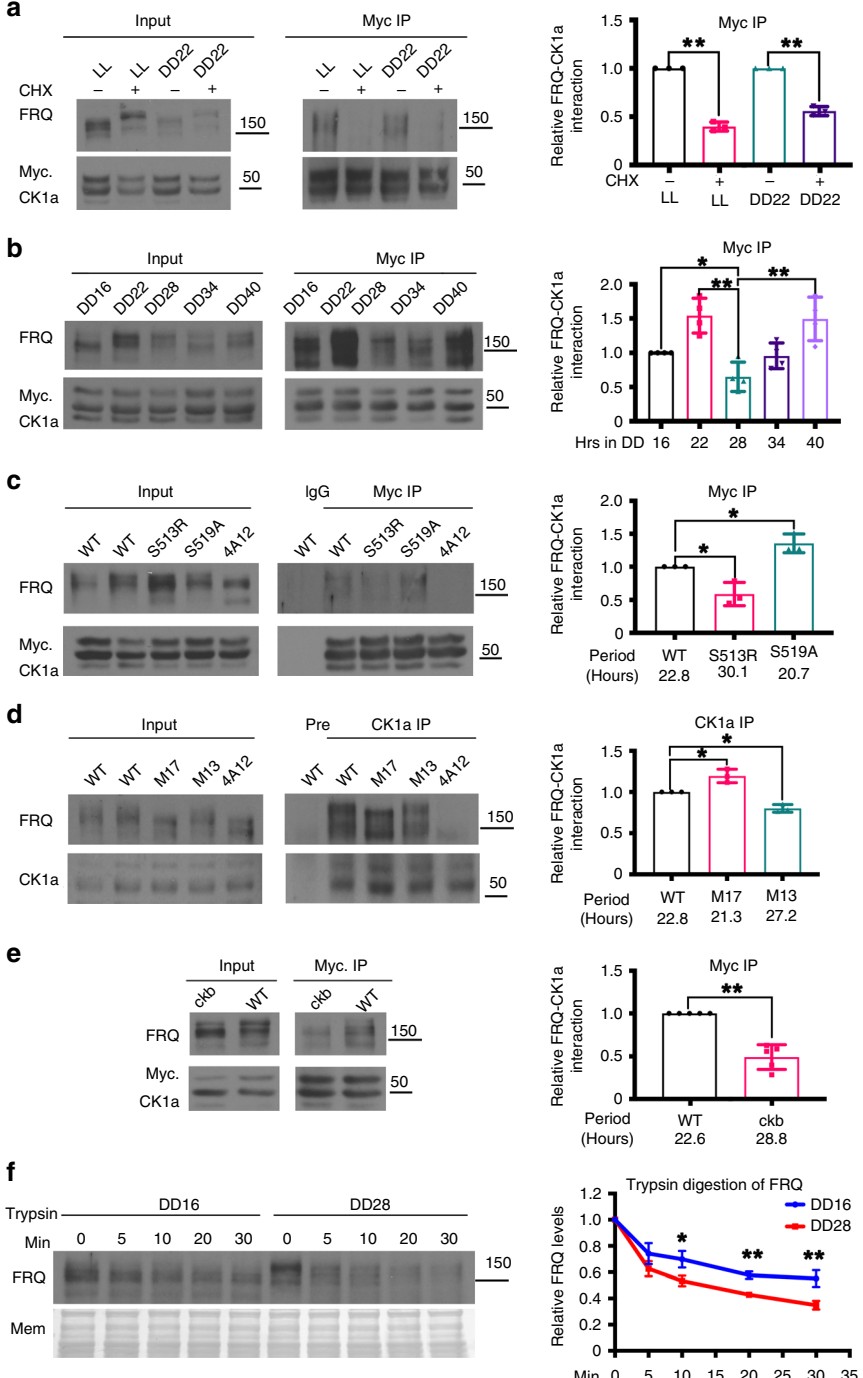

**Fig. 3** Regulation of the FRQ-CK1a interaction by FRQ phosphorylation. **a** c-Myc immunoprecipitation assay in the wild-type strain with and without CHX treatment. The cultures were grown in LL or DD and were treated with CHX for 1 h before immunoprecipitation was performed. Quantification of relative FRQ-CK1a interaction levels is based on the ratio of IP to Input. The densitometric analysis is shown on the right. Error bars are standard deviations ($n = 3$). **$P < 0.01$. Student's $t$-test was used. The position of the molecular weight marker is indicated. **b** c-Myc immunoprecipitation assay in the wild-type strain at indicated time points in DD. The densitometric analysis is shown on the right. Error bars are standard deviations ($n = 4$). *$P < 0.05$, **$P < 0.01$. Student's $t$-test was used. **c, d** c-Myc (**c**) or CK1a (**d**) immunoprecipitation assays showing the relative FRQ-CK1a interaction in the indicated FRQ phosphorylation sites mutants in LL. Long period mutants (S513R, M13) showed reduced FRQ-CK1a interaction while the interaction was increased in the short period mutants (S519A and M17). ($n = 3$). *$P < 0.05$. Student's t-test was used. **e** c-Myc immunoprecipitation assays showing the relative FRQ-CK1a interaction is decreased in the long period $ckb^{RIP}$ mutant in LL. ($n = 5$). **$P < 0.01$. Student's $t$-test was used. **f** Western blot showing trypsin digestion of hypophosphorylated (DD16) and hyperphosphorylated (DD28) FRQ. Densitometric quantification of full-length FRQ relative to the amount at time 0 is shown on the right. Error bars are standard deviations ($n = 4$). *$P < 0.05$, **$P < 0.01$. Student's $t$-test was used. Source data are provided as a Source Data file

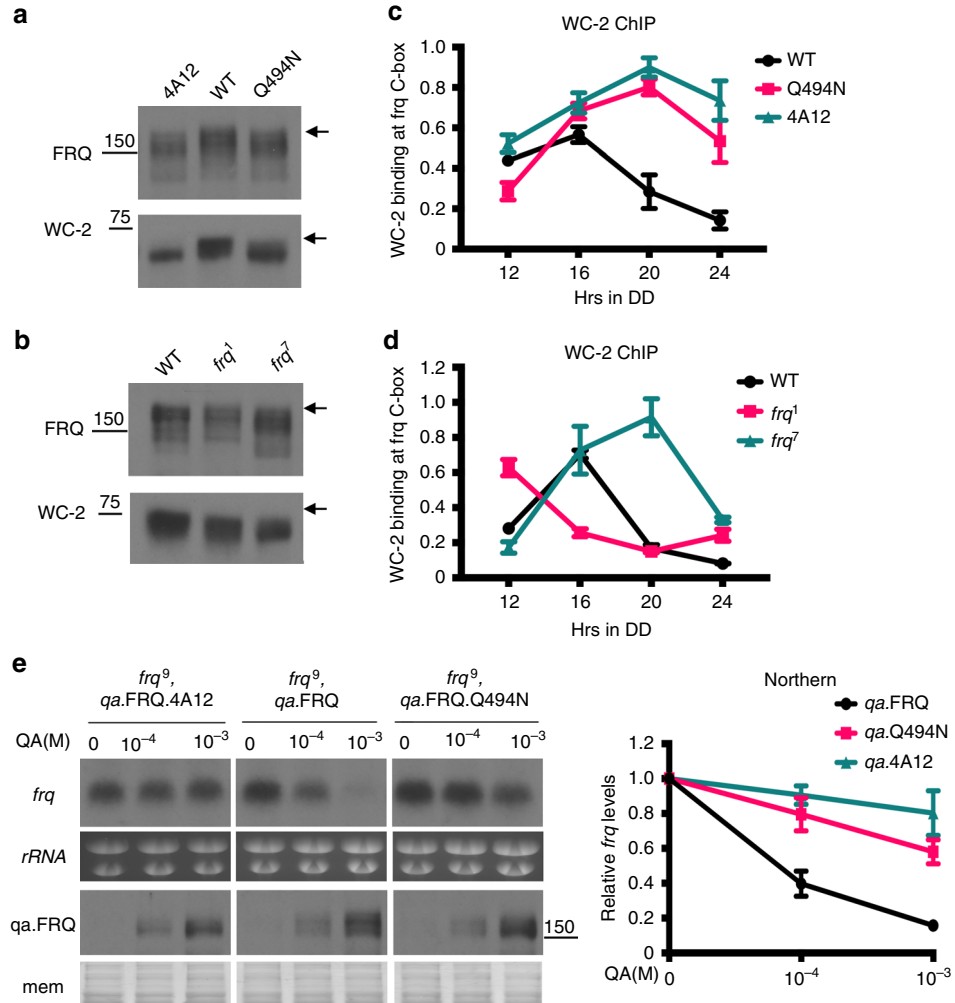

**Fig. 4** FRQ-CK1a interaction determines FRQ activity in the negative feedback loop. **a, b** Western blot analyses showing the phosphorylation profiles of FRQ and WC-2 in the indicated strains. The arrows indicate the hyperphosphorylated FRQ or WC-2 species. The position of the molecular weight marker is indicated. **c, d** WC-2 ChIP assays showing that the relative enrichment levels of WC-2 at the c-box of *frq* promoter in the indicated strains collected at different time points in DD. Error bars are standard deviations ($n = 3$). **e** Northern blot analysis showing the *frq* mRNA levels in the indicated strains after the induction of FRQ or its mutant form by QA at different concentrations. Error bars are standard deviations ($n = 3$). Source data are provided as a Source Data file

sensitivity of FRQ in *Neurospora* extracts from different time points. Hyperphosphorylated FRQ (DD28) was more sensitive to trypsin digestion than was hypophosphorylated FRQ (DD16) (Fig. 3f). This result suggests that phosphorylation induces changes of FRQ structure, which likely impacts FRQ-CK1a complex formation.

**FRQ-CK1a interaction determines FRQ activity in the feedback loop.** Our data above suggest that the FRQ-CK1a interaction influences the circadian period by determining the efficiency of FRQ phosphorylation and its stability. It was previously proposed that FRQ recruits CK1a to phosphorylate the WC proteins, which leads to the inhibition of WCC activity and *frq* transcription, closing the negative feedback loop[13,44,45,48]. This latter process should occur independently of the effect of the FRQ-CK1a interaction on FRQ stability. As expected, the 4A12 mutation, which completely abolished the FRQ-CK1a interaction, resulted in hypophosphorylation of both FRQ and WC-2 (Fig. 4a). In the Q494N mutant, which has an impaired FRQ-CK1a interaction due to the mutation in the FCD and a 30.3-h period (Fig. 2c, d), FRQ and WC-2 were hypophosphorylated

compared to the wild-type strain, but the WC-2 phosphorylation profile was intermediate between the wild-type and 4A12 strains (Fig. 4a). In addition, in the *frq*[7] strain, which also has an impaired FRQ-CK1a interaction and a long period, WC-2 was hypophosphorylated (Fig. 4b). Thus, the FRQ-CK1a interaction affects the phosphorylation efficiency of WC protein.

To determine whether the FRQ-CK1a interaction affects the circadian negative feedback process, we performed chromatin immunoprecipitation (ChIP) assays using a WC-2 antibody. The relative WC-2 enrichment at the *frq* C-box was constantly elevated in the 4A12 mutant compared to the wild-type strain at different time points in DD (Fig. 4c). As expected, intermediate levels of WC-2 enrichment at the *frq* C-box were observed in the Q494N mutant. In addition, the WC-2 enrichment was also higher in the *frq*[7] strain and lower in the *frq*[1] strain compared to the wild-type strain (Fig. 4d), consistent with previously reported FRQ levels in these mutant strains[54]. These results suggest that the FRQ-CK1a interaction determines the efficiency of WC phosphorylation by CK1a and, therefore, the DNA binding activity of WCC.

To confirm the role of the FRQ-CK1a interaction in determining FRQ activity in negative feedback process, we

compared the ability of wild-type FRQ and FCD mutants in suppressing *frq* transcription. To allow side-by-side comparison and to avoid the interference from the endogenous FRQ protein, we inducibly expressed FRQ (wild-type, 4A12 or Q494N) in a *frq*[9] strain from a construct induced by quinic acid (QA); this allowed control of FRQ expression in a QA concentration-dependent manner[33,55]. The *frq*[9] strain has a frame-shift mutation in the *frq* open reading frame that results in a truncated non-functional FRQ protein and constant high *frq* RNA level due to the loss of circadian negative feedback process[49]. FRQ protein was induced to higher levels as QA concentration increased, and the FRQ levels were similar at each QA concentration in the three *qa*-FRQ strains (Fig. 4e). Northern blot analysis was performed to detect the endogenous *frq*[9] mRNA. As expected for a normal circadian negative feedback process, the induction of wild-type FRQ expression resulted in a marked decrease of *frq*[9] mRNA level at $1 \times 10^{-4}$ M QA, and the *frq* mRNA level was very low at $1 \times 10^{-3}$ M QA (Fig. 4e). In contrast, induction of FRQ.4A12 expression, even at $1 \times 10^{-3}$ M QA, failed to repress *frq*[9] mRNA levels (Fig. 4e), indicating the complete loss of circadian negative feedback loop due to lack of FRQ-CK1a complex formation. On the other hand, the induction of FRQ.Q494N resulted in a modest decrease of *frq*[9] mRNA levels (Fig. 4e). Together, these results demonstrate that the FRQ-CK1a interaction determines FRQ activity in the circadian negative feedback loop.

Thus, the FRQ-CK1a interaction should influence the period length independent of FRQ stability by determining the time needed to close the negative feedback loop. Supporting this conclusion, we show that the mutants with stronger FRQ-CK1a interactions exhibit short period phenotypes while the mutants with weaker interactions are long period even when FRQ stability was not affected. Consistent with our findings, it was recently shown that mutations of CRY, a negative element in the mammalian circadian negative feedback loop, can result in period length changes independent of its stability in mouse cells[19].

**Mathematic modeling of the *Neurospora* circadian clock**. Our experimental data established that the physical interaction between FRQ and CK1a is a critical molecular step in determining the period of the *Neurospora* circadian clock. Previous mathematical models of the *Neurospora* circadian clock did not include CK1 or its interaction with FRQ[47,56,57]. To test the importance of physical interactions between FRQ and CK1a in silico, we constructed a mathematical model that includes explicit steps describing physical interactions between FRQ, CK1a, and WCC (Fig. 5a). Briefly, *frq* mRNA is translated ($\alpha_f$) into FRQ protein, which interacts with CK1a ($\alpha_1$). The formation of FRQ-CK1a complex leads to subsequent phosphorylation of FRQ ($\alpha_2$) followed by a physical interaction with WCC ($\alpha_3$). Additional phosphorylation of FRQ and WC proteins ($\alpha_4$) leads to the inactivation of WCC (WCC$_i$). Finally, the CK1-FRQ$_{pp}$-WCC$_i$ complex dissociates ($\alpha_5$), and each protein is either degraded or recycled. We assumed that there are two forms of WCC (WCC$_a$ and CK1-FRQ-WCC$_a$) that activate the transcription of *frq* ($\sigma_1$, $\sigma_2$), but either one alone is sufficient to generate autonomous oscillations. Each species is degraded with a rate of $\delta_{i\ (1-10)}$. In addition, we incorporated the following steps based on our experimental observations that FRQ phosphorylation has opposing roles on FRQ-CK1a interaction: (1) the initial FRQ phosphorylation events (CK1-FRQ$_p$) promote the association of FRQ and CK1a ($\sigma_3$), forming a positive feedback loop, and (2) hyperphosphorylation of FRQ (CK1-FRQ$_{pp}$) promotes the dissociation of FRQ-CK1a ($\sigma_4$), forming an additional negative feedback loop. This molecular wiring diagram was converted into

a set of ordinary differential equations for in silico analyses (see "Methods" for equations and additional information).

Our model contains 10 variables and 35 parameters. Because most of the parameter values are unknown, we performed an unbiased parameter search by using a Bayesian analysis with the Markov chain Monte Carlo estimation (Supplementary Fig. 4a±c), which generated multiple sets of default parameters that produce robust circadian oscillations of about 22 h (please see detailed description in "Methods"). Distribution of parameters from these sets of parameters are summarized as boxed plots in Supplementary Fig. 5, which indicated that each parameter has different spread with distinct mean values. We randomly chose three sets of independent parameters for in-depth sensitivity analysis to determine the sensitivity of period length to each parameter (Supplementary Tables 1, 2). To our surprise, the same top five parameters that determine the period in this system were identified with each set of parameters (Fig. 5b). These parameters can be divided into two categories: the rates that determine dynamics of the positive feedback loop that promote the association between FRQ and CK1 ($\alpha_1$, $\alpha_2$, $\alpha_c$) as the top three parameters and the rates of *frq* mRNA and FRQ$_p$-CK1a degradations ($\delta_1$, $\delta_5$). Our representative data using the parameter set 1 shows that the rate of association between FRQ and CK1a, $\alpha_1$, was the most sensitive parameter; reductions in $\alpha_1$ caused periods of more than 60 h (Fig. 5c). Reduction of the rate of CK1 synthesis ($\alpha_c$) and the rate of phosphorylation FRQ-CK1a complex ($\alpha_2$) also increased the period to more than 40 h (Fig. 5d, e).

In our model, the rate of degradation of *frq* mRNA ($\delta_1$) and FRQ degradation after phosphorylation ($\delta_5$) moderately influenced the period, up to 36 h in case of $\delta_1$ and only 27 h for $\delta_5$ (Fig. 5f, g). In contrast, most of the other degradation rates, including the rates of degradation of hyperphosphorylated FRQ ($\delta_8$ and $\delta_9$), showed a large region of oscillatory domain with minimal changes in period (Fig. 5h and Supplementary Fig. 6). This is consistent with the fact that the hyperphosphorylated FRQ inhibits the FRQ-CK1a interaction, and thus it is not as functional as hypophosphorylated FRQ in the negative feedback loop. Thus, these in silico experiments suggest that FRQ degradation only has a modest role in period determination.

The dramatic period effect due to reduction of the rate of association between FRQ and CK1a ($\alpha_1$) occurs because the system is approaching to a bifurcation point (i.e., a point in parameter space where the stability of a system changes altering the behavior of the system) at which an unstable steady state with a stable limit cycle collides with a saddle point (called a Saddle Node on Invariant Circle)[58,59], generating an oscillation with an infinite period (Fig. 5i). Biologically, the observed dramatic increase of period occurs when any of the rates that determine the accumulation of CK1-FRQ$_p$ ($\alpha_1$, $\alpha_2$, $\alpha_c$), which triggers positive feedback loop, are very low, resulting in a critical slowdown of the system. In other words, a certain amount of CK1-FRQ$_p$ has to accumulate to trigger the positive feedback loop promoting the association of FRQ and CK1a. Therefore, if $\alpha_1$, $\alpha_2$, or $\alpha_c$ are low, then it takes much longer time to accumulate a critical concentration of CK1-FRQ$_p$ that will induce the positive feedback loop. In contrast, this unique dynamical behavior is not observed with other parameters in the model. Bifurcation analysis of $\delta_1$ showed that the oscillatory domain is bounded by two Hopf bifurcation points (Fig. 5j, solid green circles) with modest monotonical decrease of period with increasing $\delta_1$[58,59]. Representative one-parameter bifurcation diagrams for $\delta_1$ and $\delta_8$ are shown for comparison with that of $\alpha_1$ in Fig. 5j, k.

The level of CK1 is known to be relatively constant, and the phosphorylation of FRQ by CK1a ($\alpha_2$) depends on the prior interaction between these two proteins ($\alpha_1$). Therefore, our

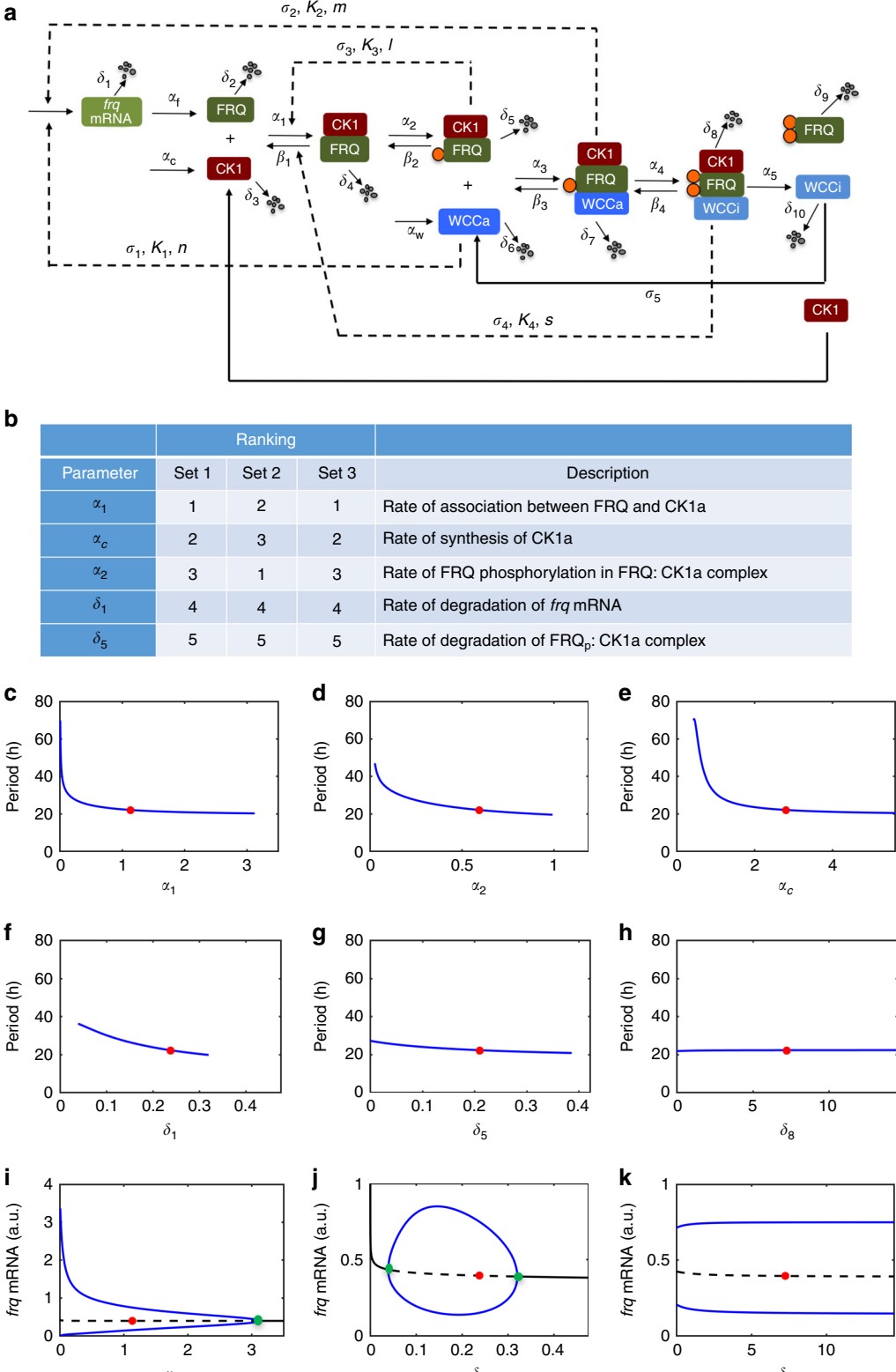

**Fig. 5** Mathematical modeling of circadian period determination. **a** Wiring diagram for the *Neurospora* circadian clock. Dashed arrows indicate modes of action of proteins or protein complexes, and the letters above each arrow indicate parameters involved in the step. Solid orange circles indicate phosphorylation of FRQ by CK1a. Please see the methods and supplementary information for equations and description of parameters. **b** The top five parameters that influence the period of the *Neurospora* circadian clock identified from computer simulations using three independent sets of parameters. **c–h** Representative changes of period as a function of $\alpha_1$, $\alpha_2$, $\alpha_c$, $\delta_1$, $\delta_5$, and $\delta_8$ from the parameter set 1. Solid red circles indicate default parameter values. **i–k** One-parameter bifurcation diagrams as functions of $\alpha_1$, $\delta_1$, and $\delta_8$ indicating the steady state of *frq* mRNA. Solid black lines represent stable steady state and dashed black lines represent unstable steady state with stable limit cycle. Blue lines indicate maximum/peak and minimum/trough of *frq* mRNA oscillations. Solid red circles indicate default parameter values, and solid green circles indicate Hopf bifurcation points

computational analyses revealed that the regulated complex formation between FRQ and CK1a is the major parameter in determining the period of the *Neurospora* circadian clock. In addition, our in silico analyses suggest that the molecular wiring diagram is one of the critical constraints that determine appropriate parameter space for robust oscillations.

## Discussion

The approximately 24-h periodicity of circadian clocks allows organisms to adapt to daily changes of environmental conditions. Protein stability was previously proposed to be the main determinant of circadian period in eukaryotes. This was challenged by a recent study in *Neurospora* that demonstrated that period length and FRQ stability could be uncoupled in *fwd-1* mutants as well as WT stains[23]. Larrondo et al. previously suggested that daily phosphorylation but not protein stability as the key factor in period determination. In this study, we established that the physical interaction between FRQ and CK1a is a major molecular process determining the circadian period in *Neurospora*.

The importance of the FRQ-CK1a interaction in period determination can be explained by its two distinct roles in the core circadian negative feedback loop. First, the FRQ-CK1a interaction determines FRQ phosphorylation efficiency. Although FRQ phosphorylation efficiency is influenced by CK1a kinase activity and by the amount of CK1a, CK1a can only phosphorylate FRQ after the two proteins form a tight complex. Thus, the binding of FRQ to CK1a is the most critical determinant for FRQ phosphorylation efficiency[13,38,39]. FRQ phosphorylation regulates both FRQ stability and the binding between FRQ and CK1a. Second, FRQ acts as the scaffold between CK1a and WCC and the rhythmic binding between FRQ and CK1a results in rhythmic WC phosphorylation by CK1a. Therefore, the FRQ-CK1a interaction also determines the efficiency of WC phosphorylation by CK1a. This process defines the time needed to close the circadian negative feedback loop. A weakened FRQ-CK1a interaction results in impaired WC phosphorylation, higher WC binding at the *frq* promoter, higher *frq* transcription and long period phenotypes.

The feedback of FRQ phosphorylation to the formation of the FRQ-CK1a complex indicates that FRQ proteins with different phosphorylation profiles are functionally distinct. The daily rhythm observed in the FRQ phosphorylation profile results in a robust daily oscillation of FRQ activity (Fig. 3). Although FRQ stability affects the amount of functional FRQ available, the interaction between FRQ and CK1a determines FRQ phosphorylation and FRQ activity in the circadian negative feedback process. Therefore, the FRQ-CK1a interaction can influence the period length independently of FRQ stability.

The period phenotypes of many *Neurospora* clock mutants were previously attributed to differences in FRQ stability[9,14,15,50]. As we showed here, these mutants also have altered amounts of FRQ-CK1a complex, which negatively correlates with their period lengths (Figs. 1 and 3). In addition, some of these period mutants exhibited altered FRQ-CK1a interaction but normal FRQ stability (Fig. 1), indicating that the FRQ-CK1a interaction can determine period length independently of the efficiency of FRQ degradation. Furthermore, point mutations that specifically altered the FRQ-CK1a interaction resulted in the most severe period mutants ever reported (Fig. 2). Finally, the stability of hyperphosphorylated FRQ does not have a major role in period determination because hyperphosphorylated FRQ has reduced ability to form a complex with CK1a (Fig. 3a, b). Together, these results indicate that FRQ-CK1a interaction, but not FRQ stability, is the major determinant of circadian period length in *Neurospora*. This conclusion is further supported by the results of our mathematical modeling

and computer simulations. Despite the critical role of the FRQ-CK1a interaction in period determination, other processes in the circadian feedback loops, such as *frq* mRNA synthesis/degradation rate, FRQ phosphorylation events affects the interaction between FRQ-WCC, WC levels and WC phosphorylation independent of FRQ, can also contribute to period determination.

The formation of a tight stoichiometric complex between clock proteins and CK1 is conserved in *Drosophila* and mammalian circadian mechanisms[12,14,24–26]. In *Drosophila*, deletion of the Doubletime (DBT) interaction domain of PER abolishes the ability of PER to be phosphorylated by CK1 and severely impairs the function of PER in transcriptional suppression[60,61]. In addition, DBT is required for the PER-dependent phosphorylation of CLOCK (CLK), which is associated with repression of E-box-dependent transcription[62,63]. The PER-CK1 interaction domain appears to be conserved in mammals[27,29,64]. In addition, the overexpression of the CK1δ/ε binding domain of PER2 abolishes circadian rhythms in cells[65], indicating the critical importance of PER-CK δ/ε binding in clock function. Moreover, CK1δ phosphorylation of CLOCK-BMAL1 in the complex was recently suggested to promote the dissociation of CLOCK-BMAL1 from DNA[26]. These results suggest that, as in *Neurospora*, the complex formation between PER and CK1 in *Drosophila* and *mammals* regulates both PER phosphorylation and the circadian negative feedback process, and thus the PER-CK1 interaction may have a similar role in period determination in animal circadian systems.

## Methods

**Strains, culture conditions, and race tube assays**. The 87–3 (*bd*, a) strain or a *frq* complementation strain (KAJ120) was used as the wild-type strain in this study. S513R, S519A, M9, M10, M13, M17, and *ckb*^RIP strains were created previously[9,15,35]. The *frq*^KO (*bd*, *his-3*) strains were used as host strains for constructs with his-3-targeting vectors. The control strain was a *frq*^KO strain transformed with KAJ120, a vector containing the wild-type *frq* gene. All constructs bearing FCD1 and FCD2 mutations were transformed into the *frq*^KO strain. Constructs with the *qa-2* promoter driving expression of Myc.His.CK1a and pBT6 were co-introduced into the KAJ120, S513R, S519A, and 4A12 strains by homologous recombination. Positive transformants were verified by western blot analyses. Constructs with the *qa-2* promoter driving expression of Myc.His.CK1a.hph were introduced into the 87–3, *frq*^1, *frq*^7, and *ckb*^RIP strains at the *csr* locus by homologous recombination. Constructs with the *qa-2* promoter driving expression of Myc.His.CK1a.bar were introduced into the KAJ120, V320I and Q494N strains at the *csr* locus by homologous recombination. pBA50 constructs with the *qa-2* promoter driving expression of WT, Q494N, or 4A12 was introduced into the *frq*^9 (*bd*, *his-3*) strain at the *his-3* locus by homologous recombination. Positive transformants were identified by western blot analyses, and homokaryon strains were isolated by microconidia purification using 5-μm filters (Millipore, Cat#SLSV025LS).

Liquid cultures were grown in minimal media (1× Vogel's, 2% glucose). When QA (Sigma-Aldrich, Cat#138622-100 G) was used to activate the *qa-2* promoter, liquid cultures were grown in $1 \times 10^{-2}$ M QA (pH 5.8), 1× Vogel's, 0.1% glucose, and 0.17% arginine. For rhythmic experiments, *Neurospora* was cultured in petri dishes in liquid medium for 2 days. The *Neurospora* mats were cut into discs and transferred into medium-containing flasks and were harvested at the indicated time points.

The medium for race tube assays contained 1× Vogel's salts, 0.1 or 0.03% glucose, 0.17% arginine, 50 ng/mL biotin (Sigma-Aldrich, Cat#B4639-500MG), and 1.5% agar with or without QA. After entrainment of 24 h in the LL condition, race tubes were transferred to the DD conditions and marked every 24 h.

**Protein and RNA analyses**. Protein extraction, quantification, and western blot analyses were performed as previously described[51,52,66]. Briefly, tissue was ground in liquid nitrogen with a mortar and pestle and suspended in ice-cold extraction buffer (50 mM HEPES (pH7.4), 137 mM NaCl, 10% Glycerol) with protease inhibitors at final concentrations: Pepstatin A (1 μg/mL), Leupeptin (1 μg/ml), PMSF (1 mM). For certain experiments with PPase inhibitors (made fresh): 25 mM NaF, 10 mM $Na_4P_2O_7.10H_2O$, 2 mM $Na_3VO_4$, 1 mM EDTA. After centrifugation, protein concentration was measured using protein assay dye (Bio-Rad, Cat#500-0006). For western blot analyses, equal amounts of total protein (40 μg) were loaded in each protein lane. After electrophoresis, proteins were transferred onto PVDF membrane (Immobilon-P, EMD Millipore, Cat#IPV1100010), and western blot analyses were performed. To analyze the phosphorylation profile of WC-2, 7.5% SDS-PAGE gels containing a ratio of 149:1 acrylamide/bisacrylamide were

used. Otherwise, 7.5% SDS-PAGE gels containing a ratio of 37.5:1 acrylamide/bisacrylamide were used.

RNA was extracted with Trizol (Ambion, Cat#15596018) in accordance with the manufacturer's protocol, and then further purified with 2.5 M LiCl (Sigma-Aldrich, L4408-500G)[67]. For northern blot analyses, equal amounts of total RNA (20 µg) were loaded onto agarose gels. After electrophoresis, the RNA was transferred onto nitrocellulose membrane (GVS North America, Cat#1212590). The membrane was probed with [$^{32}$P] UTP (PerkinElmer, Cat#BLU007H250UC)-labeled RNA probes specific for *frq* 5′UTR. RNA probes were transcribed in vitro from PCR products by T7 RNA polymerase (Ambion) as per the manufacturer's protocol. The *frq* primer sequences used for the template amplification were *frq* Northern F, 5′-TAATACGACTCACTATAGGGGAAGCCTGCGATGCTACTC AC-3′, and *frq* Northern R, 5′-CATCGCATCCACTTCGCACAC-3′. The list of primers used in this study is shown in Supplementary Table 3.

**Immunoprecipitation and FLAG pull-down analyses.** IP analyses were performed as previously described[51]. Briefly, *Neurospora* proteins were extracted as described above. For each immunoprecipitation reaction, 2 mg protein and 1.5 µL c-Myc antibody (Roche, Cat#11667203001), 1 µL WC-2 antibody[43], or 1 µL CK1a antibody were used. After incubation with antibody for 3 h, 40 µL GammaBind G Sepharose beads (GE Healthcare) were added, and samples were incubated for 1 h. For the FLAG pull-down assay, 4 mg purified FLAG-CK1a fusion protein and 2 µL FLAG antibody (Sigma-Aldrich, Cat#F1804) were incubated with 2 mg *Neurospora* extracts. After incubation with antibody for 3 h, 40 protein GammaBind G Sepharose beads were added, and samples were incubated for 1 h. Immunoprecipitated proteins were washed three times using extraction buffer before Western blot analysis. IP experiments were performed using cultures harvested in constant light (LL) except for the rhythmic experiments. Quantification of relative FRQ-CK1a or FRQ-WC-2 interaction levels are based on the ratio of IP to Input and normalized with CK1a or WC-2 levels, respectively.

**Protein stability assays.** For CHX (Sigma-Aldrich, Cat#C7698-5G) treatments, discs made from 2-day-old mycelial mats of individual strains were grown in liquid media for 1 day under constant light conditions. CHX was then added to a final concentration of 10 µg/mL, and tissues were harvested at the indicated time points.

**Trypsin digestion assays.** For the trypsin-sensitivity assay, protein extracts were diluted to a protein concentration of 2.5 µg/µL. A 100-µL aliquot was treated with trypsin (final concentration 1 µg/mL) at 25 °C. A 20 µL sample was taken from the reaction at each time point (0, 5, 15, and 30 min) after addition of trypsin. Protein samples were mixed with protein loading buffer and resolved by SDS-PAGE. To compare trypsin sensitivity of FRQ from different strains, experiments were performed side by side, and the protein samples were transferred to the same membrane for western blot analysis.

**ChIP analyses.** ChIP assays were performed as previously described[48,68]. Briefly, *Neurospora* tissues were fixed with 1% formaldehyde for 15 min at room temperature with shaking. Glycine was then added at a final concentration of 125 mM. The cross-linked tissues were ground and resuspended in lysis buffer (50 mM HEPES, pH 7.5, 137 mM NaCl, 1 mM EDTA, 1% Triton X-100, 0.1% deoxycholate, 0.1% SDS). Chromatin was sheared using sonication to 500–1000 base-pair fragments. For each immunoprecipitation reaction, 1 mg protein was used. WC-2 antibody was generated previously[43]. The ChIP reaction was carried out with 2 µL WC-2 antibody. Immunoprecipitated DNA was enriched using GammaBind G Sepharose beads and washed sequentially with low-salt washing buffer (0.1% SDS, 1% Triton X-100, 2 mM EDTA, 150 mM NaCl, 20 mM Tris–HCl, pH 8), high-salt washing buffer (0.1% SDS, 1% Triton X-100, 2 mM EDTA, 500 mM NaCl, 20 mM Tris–HCl, pH 8), LNDET buffer (0.25 M LiCl, 1% NP40, 0.1% deoxycholate, 1 mM EDTA,10 mM Tris–HCl, pH 8), and TE buffer (1 mM EDTA, 10 mM Tris–HCl, pH 8). Elution buffer (1% SDS, 0.1 M NaHCO3) was used to retrieve protein-DNA complex. Finally, eluted DNA was purified and quantified by real-time qPCR. The primer sets used were *frq* C-box F, 5′-GTCAAGCTCGTACCCACATC-3′, and *frq* C-box R, 5′-CCGAAAGTATCTTGAGCCTCC-3′. Occupancies were normalized by the ratio of ChIP to Input DNA.

**Generation of CK1a antibody.** The GST-CK1a fusion protein (containing CK1a amino acids Q299-E384) was expressed in BL21 cells, and the recombinant proteins were purified and used as the antigens to generate rabbit polyclonal antisera[69].

**Quantification and statistical analyses.** Quantification of western blot and northern blot data were performed using Image J software. All studies were performed at least three independent experiments (different days and different preparations). Error bars are standard deviations for IP assays and standard error of means for race tube assays. Statistical significance was determined by Student's *t*-test.

**Mathematical model.** Variables used are:

$[F_m]$ = concentration of *frq* mRNA
$[F]$ = concentration of FRQ
$[C]$ = concentration of CK1
$[F{:}C]$ = concentration of complex FRQ:CK1
$[F_p{:}C]$ = concentration of complex FRQ$_p$:CK1
$[W_a]$ = concentration of WCC$_a$
$[W_i]$ = concentration of WCC$_i$
$[F_p{:}C{:}W_a]$ = concentration of complex FRQ$_p$:CK1:WCC$_a$
$[F_{pp}{:}C{:}W_i]$ = concentration of complex FRQ$_{pp}$:CK1:WCC$_i$
$[F_{pp}]$ = concentration of double-phosphorylated FRQ
Parameters used are listed below:
$\alpha_1, \alpha_2, \alpha_3, \alpha_4$: forward reaction rates
$\alpha_5$: dissociation rate
$\beta_1, \beta_2, \beta_3, \beta_4$: backward reaction rates
$\alpha_f$: synthesis rate of FRQ protein
$\alpha_c$: synthesis rate of CK1
$\alpha_w$: synthesis rate of WWCa
$\delta_1, \delta_2, \delta_3, \delta_4, \delta_5, \delta_6, \delta_7, \delta_8, \delta_9, \delta_{10}$: degradation rates
$\sigma_1, \sigma_2$: transcription rates
$\sigma_3$: activation rate of autocatalysis
$\sigma_4$: F$_{pp}$:C:W$_i$ dependent dissociation of F:C
$\sigma_5$: conversion of W$_i$ to W$_a$
$K_1, K_2, K_3, K_4$: thresholds
$m, n, \ell, s$: Hill coefficients
System of ordinary differential equations are listed below:

$$\frac{d[F_m]}{dt} = \sigma_1 \frac{[W_a]^m}{K_1^m + [W_a]^m} + \sigma_2 \frac{[F_p{:}C{:}W_a]^n}{K_2^n + [F_p{:}C{:}W_a]^n} - \delta_1[F_m] \quad (1)$$

$$\frac{d[F]}{dt} = \alpha_f[F_m] - \alpha_1[F][C] + \beta_1[F{:}C] - \sigma_3 \frac{[F][C][F_p{:}C]^\ell}{K_3^\ell + [F_p{:}C]^\ell} + \sigma_4[F{:}C]\frac{[F_{pp}{:}C{:}W_i]^s}{K_4^s + [F_{pp}{:}C{:}W_i]^s} - \delta_2[F] \quad (2)$$

$$\frac{d[C]}{dt} = \alpha_c - \alpha_1[F][C] + \beta_1[F{:}C] - \sigma_3\frac{[F][C][F_p{:}C]^\ell}{K_3^\ell + [F_p{:}C]^\ell} + \alpha_5[F_{pp}{:}C{:}W_i] + \sigma_4[F{:}C]\frac{[F_{pp}{:}C{:}W_i]^s}{K_4^s + [F_{pp}{:}C{:}W_i]^s} - \delta_3[C] \quad (3)$$

$$\frac{d[F{:}C]}{dt} = \alpha_1[F][C] - \beta_1[F{:}C] + \sigma_3\frac{[F][C][F_p{:}C]^\ell}{K_3^\ell + [F_p{:}C]^\ell} - \sigma_4[F{:}C]\frac{[F_{pp}{:}C{:}W_i]^s}{K_4^s + [F_{pp}{:}C{:}W_i]^s} - \alpha_2[F{:}C] + \beta_2[F_p{:}C] - \delta_4[F{:}C]$$
$$= \alpha_1[F][C] + \sigma_3\frac{[F][C][F_p{:}C]^\ell}{K_3^\ell + [F_p{:}C]^\ell} - \sigma_4[F{:}C]\frac{[F_{pp}{:}C{:}W_i]^s}{K_4^s + [F_{pp}{:}C{:}W_i]^s} + \beta_2[F_p{:}C] - (\alpha_2 + \beta_1 + \delta_4)[F{:}C] \quad (4)$$

$$\frac{d[F_p{:}C]}{dt} = \alpha_2[F{:}C] - \beta_2[F_p{:}C] - \alpha_3[F_p{:}C][W_a] + \beta_3[F_p{:}C{:}W_a] - \delta_5[F_p{:}C]$$
$$= \alpha_2[F{:}C] - \alpha_3[F_p{:}C][W_a] + \beta_3[F_p{:}C{:}W_a] - (\beta_2 + \delta_5)[F_p{:}C] \quad (5)$$

$$\frac{d[W_a]}{dt} = \alpha_w - \alpha_3[F_p{:}C][W_a] + \beta_3[F_p{:}C{:}W_a] + \sigma_5[W_i] - \delta_6[W_a] \quad (6)$$

$$\frac{d[F_p{:}C{:}W_a]}{dt} = \alpha_3[F_p{:}C][W_a] - \beta_3[F_p{:}C{:}W_a] - \alpha_4[F_p{:}C{:}W_a] + \beta_4[F_{pp}{:}C{:}W_i] - \delta_7[F_p{:}C{:}W_a]$$
$$= \alpha_3[F_p{:}C][W_a] + \beta_4[F_{pp}{:}C{:}W_i] - (\beta_3 + \alpha_4 + \delta_7)[F_p{:}C{:}W_a] \quad (7)$$

$$\frac{d[F_{pp}{:}C{:}W_i]}{dt} = \alpha_4[F_p{:}C{:}W_a] - \beta_4[F_{pp}{:}C{:}W_i] - \alpha_5[F_{pp}{:}C{:}W_i] - \delta_8[F_{pp}{:}C{:}W_i]$$
$$= \alpha_4[F_p{:}C{:}W_a] - (\beta_4 + \alpha_5 + \delta_8)[F_{pp}{:}C{:}W_i] \quad (8)$$

$$\frac{d[F_{pp}]}{dt} = \alpha_5[F_{pp}{:}C{:}W_i] - \delta_9[F_{pp}] \quad (9)$$

$$\frac{d[W_i]}{dt} = \alpha_5[F_{pp}{:}C{:}W_i] - \sigma_5[W_i] - \delta_{10}[W_i] = \alpha_5[F_{pp}{:}C{:}W_i] - (\sigma_5 + \delta_{10})[W_i] \quad (10)$$

As a positive control, we measure the *frq* gene expression using a luciferase reporter fused with the *frq* promoter in constant dark (DD) conditions in three independent experiments (Supplementary Fig. 4a). The independent experiments result in different *frq* expression levels, each with slightly increasing trend. To fit our mathematical model assuming a periodic curve of *frq*, we removed the estimated trend from the original expression level by using a second-order polynomial regression, then standardized the detrended series (Supplementary Fig. 4b).

To estimate the model parameters in the above mathematical model, we adopt a Bayesian analysis with the Markov chain Monte Carlo (MCMC) estimation. In the statistical inference, we view the experimental measurement of *frq* as a realization of the underlying biological process. Let $\tilde{y}_{it}$ denote the detrended and standardized expression level measured at time $t$ in the $i$-th experiment, $i = 1, 2, 3$ and $t = 56, \ldots, 121$ (Supplementary Fig. 4b). The simulation model output for time $t$ is denoted by $y_t$ and its standardized value as $\tilde{y}_t$, i.e., $\tilde{y}_t = (y_t - \bar{y})/s_y$ where $\bar{y}$ and $s_y$ are the sample mean and standard deviation of the model outputs $\{y_t\}$, respectively.

We put a simple assumption that the deviation of $\tilde{y}_{it}$ from the simulation output follows the Gaussian random error, i.e., $\tilde{y}_{it} = \bar{y}_t + \varepsilon_t$ where $\varepsilon_t \sim N(0, \sigma^2)$. To restrict the solution to be a periodic curve, we put the informative prior distribution: $\bar{y}_t - \bar{y}_{t+22} \sim N(0, \tau^2)$ for $56 \leq t \leq 99$. We set the tuning parameter (hyperparameter) $\tau$ as 0.1, so that the fitted values of $\{\bar{y}_t\}$ do not result in a nonperiodic curve. For other parameters, we put weakly informative priors, so the parameter estimation is mainly driven by the experimental data rather than the prior distribution. Then, we run MCMC to explore the region of the candidate parameter sets whose simulation output $\{\bar{y}_t\}$ is similar to the experimental observation $\{\tilde{y}_{it}\}$, $i = 1, 2, 3$. After 1000 iterations of burn-in period, we store the MCMC draws up to 4000 iterations, resulting in numerous sets of model parameter estimates. Distribution of parameters is summarized in Supplementary Fig. 5 as boxed plots with a horizontal black line indicating the mean of each parameter. We observe that each parameter has different spread with distinct mean values. For example, $\alpha_5$ shows a large spread of parameter space ranging from close to 0 to about 11 with a mean value close to 0. This indicates that $\alpha_5$ is not a critical parameter in the system, because oscillations can be generated even when its value is close to 0 (with a wide range of values of $\alpha_5$). In contrast, $\delta_1$ and $\delta_5$ show tighter region of parameter space, which suggest that they may play critical roles in generating autonomous oscillations. However, distribution of these parameters does not indicate how the circadian period is altered as a function of each parameter.

Supplementary Fig. 4c shows that the model-fitted *frq* gene expression levels (solid curves) determined by the randomly selected three different sets of estimated parameters are similar to each other. Supplementary Table 1 shows that the parameter values are distinct which show uncertainty in the parameter estimation. In other words, there exist multiple sets of parameters whose simulation outputs (the *frq* curves) are identical.

**Bifurcation and sensitivity analyses**. For each parameter set, we performed one-parameter bifurcation analyses to investigate the dynamical changes in the *Neurospora* circadian clock system such as changes in period and molecular levels. Bifurcation diagrams were obtained by varying a single parameter, while the rest of the parameters were held fixed as their default values; see Fig. 5c–k and Supplementary Fig. 6.

We also evaluated the period sensitivity of each parameter for three parameter sets as follows:

$$\phi = \frac{\Delta P}{P_0}$$

where $P_0$ is the period of default parameter values and $\Delta P$ is the difference between the maximum period and minimum period as each parameter varies from 0 to 100% more than the default parameter value. For each parameter set, the ranking of period sensitivity is listed in Supplementary Table 2.

**Reporting summary**. Further information on research design is available in the Nature Research Reporting Summary linked to this article.

## Data Availability
The source data underlying Figs. 1b–f, 2e, f, 3a–f, 4a, b, 4e, and Supplementary Figs 1a, 1c, d, 2a–b, 3a, and 3c-d are provided as a Source Data file. The authors declare that all other data supporting the findings of this study are available within the article and its Supplementary Information files, or from the corresponding author upon request.

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

## Acknowledgements

We thank members of our laboratories for technical assistance and Dr. Jay Dunlap for suggestions. This work was supported by grants from National Institutes of Health (R35 GM118118), Cancer Prevention and Research Institute of Texas (RP160268), and the Welch Foundation (I-1560) to Y.L., Department of Interior (D12AP00005) to C.I.H. and S.L., State Key Program of National Natural Science of China (31330004) and the National Basic Research Program of China (973 Program) (2012CB947600) to Q.H., and CAS Pioneer Hundred Talents Program to X.L. A. Caicedo-Casso was supported by Universidad del Valle via the research project CI 71033.

## Author contributions

Conceptualization, X.L., and Y.L.; methodology, X.L., A.Chen, C.H., Q.H., A. Caicedo-Casso, S.L, H.J.K., and Y.L.; investigation, X.L., A. Chen, C.H., A. Caicedo-Casso, S.L., H.J.K., G.C., and M.D.; data analysis, X.L., A. Chen, C.H., A. Caicedo-Casso, S.L., H.J.K., Q.H., and Y.L.; writing, original draft, X.L. and Y.L.; writing, review & editing, X.L., C.H., and Y.L.; funding acquisition, X.L., C.H., Q.H., and Y.L.; resources, X.L., A. Chen, C.H., Q.H., and Y.L.; supervision, Y.L.

## Additional information

**Competing interests:** The authors declare no competing interests.

