## [Peer Review File · Nature Communications]

Reviewers' comments:

Reviewer #1 (Remarks to the Author):

The authors present an intriguing hypothesis--that the stability of the CK1-FRQ complex in *Neurospora* (and potentially in other eukaryotic clocks) is a key determinant of period length. They provide a set of biochemical analyses and then a mathematical model in support of this hypothesis. Some of the results are quite compelling, including the generation of very long period alleles by rationally designing mutations in FRQ to disrupt interaction with CK1. However I feel that there are some serious weaknesses in the current version of this manuscript which make the results hard to interpret.

1. The hypothesis seems to predict (and this is supported by the authors' interpretation of their mathematical model) that instability of the CK1-FRQ complex will result in slowed accumulation of FRQ phosphorylation. When the complex is very weak, the period will lengthen dramatically because it is difficult to trip the positive feedback loop that enhances CK1-FRQ affinity. Thus a strong prediction is that the phosphorylation phase of the clock cycle will be specifically extended in mutants that decrease CK1-FRQ stability. This seems central to the study and should be tested directly.

2. It is unclear what time points in the clock cycle were used for the interaction measurements e.g. in the IPs in Figure 1. Since the interaction between CK1 and FRQ is rhythmic, this should be clarified so we can be sure it is a fair comparison between different mutants which may show peak interaction at different times.

3. The mathematical modeling suggests that changes in CK1 specific activity in the complex can also have dramatic effects on period. This is consistent with the idea that what really matters is the rate at which FRQ phosphorylation can accumulate. Is this inconsistent with the authors' conclusions? The text seems to focus on the parameters that determine complex stability.

minor points:

1. I'm not sure why the sensitivity analysis is limited to 3 parameter sets taken from the MCMC distribution. Surely it would not be too computationally intensive to calculate a distribution of sensitivities for everything that fit well.

2. One potentially interesting prediction from this study is that mutants with decreased FRQ-CK1 stability may be more sensitive than wildtype to CK1 over expression (since this could drive the binding equilibrium towards complex formation)

Reviewer #2 (Remarks to the Author):

The manuscript "Formation of the FREQUENCY-CK1 complex underlies circadian period determination" by Liu et al. is a worthy addition to our understanding of the molecular daily clock in *Neurospora*, with potentially important ramifications to the mammalian and fly clock mechanisms. However, I have two concerns about the supporting evidence for the hypothesis presented in this manuscript:

1. The support for the hypothesis is basically the same kind of correlative support upon which the previous hypothesis (i.e., FRQ degradation rate) was based. In other words, previously the authors correlated period with FRQ degradation and claimed this correlation supported a causal link that FRQ degradation rate determined the circadian period. Now the authors say that this correlation breaks down with further examples, and their new hypothesis is that the FRQ/CK1 interaction is

the cause. But the data are again correlative, not analytical. The authors set up the argument as though EITHER FRQ degradation rate is the cause OR the strength of FRQ/CK1 interaction is the cause. What about another possibility?

These correlations are largely based on mutants that are already known to have period defects, and there is only a little work with mutants that were first identified to alter FRQ/CK1 interaction to “hypothesis test” their ideas. (e.g., mutations in the FCD domains caused long period, but unfortunately there are no mutations in either FCD1 or FCD2 that can enhance the FRQ-CK1a interaction and cause shorter period phenotypes, which would be a persuasive demonstration.) Moreover, are there any mutants that affect FRQ/CK1 interaction that do NOT cause period defects? Frankly, I think their presentation would be enhanced if they reorganize to emphasize hypothesis testing with previously unreported mutations in the FRQ/CK1 interaction domains to test whether those interaction domain mutants also exhibit period effects. This would be a more persuasive approach to present the hypothesis and data.

2. What does “interaction” mean to these authors? Isn’t “affinity” and/or “binding constant” the relevant parameter? Is the relative amount of immunoprecipitated protein really the same thing as “strength” of interaction? The chronobiology field is moving to more and more rigorous biochemical criteria, and the amount of immunoprecipitated protein just doesn’t seem to be a rigorous demonstration of affinity. How do they calculate and normalize the interaction? The Fig. 1 legend states “Quantification of relative FRQ-WC-2 interaction levels is based on the ratio of IP to Input.” How do the authors correct for loading errors and sample-to-sample variability of IP efficiency?

Can the authors purify these proteins and form a complex in vitro? How about testing interaction of native vs. mutant proteins where one protein is immobilized on a Biacore surface and the interaction is tested by SPR? Perhaps CK1 could be immobilized and the IP-complex flowed over the surface and an affinity SPR measurement performed? (I recognize that the complex might not bind to CK1 on the surface because CK1 in the complex is masking the interaction sites, but perhaps the authors can think of a better way to accomplish the same goal.)

Minor Points/Suggestions:

Fig. 2d: The correlation between “interaction” (ordinate) and period (abscissa) is rather catastrophic (exponential); there aren’t enough intermediate values of period to make this correlation persuasive.

Fig. 3b (right): why does the period seem to be so short?

Fig. 3c/d: only a few mutants are shown here. Don’t you have more mutants available to test these correlations? The differences are small and/or variable and not very persuasive.

Fig. 4a/b vs. 4c/d: the effects of the mutations on phosphorylation (a/b) are small and subtle, whereas the ChIP effects (c/d) appear to be quite large. It does not appear that these two effects are directly correlated.

Fig. 5: why has modeling only just now revealed that the rate of FRQ degradation minimally affects period? What is different about this model and why did previous models indicate that FRQ degradation could/would have a significant effect on period? Other than the current study that precipitated this new model, are there any particularly designed experiments that support the proposed mathematical modeling and computational simulations for the FRQ-CK1a interaction and period determination?

DISCUSSION: The authors should address in the Discussion possible alternative interpretations of their data. Also, did the authors test any of the fwd-1 mutants (from Larrondo et al. 2015) to

determine if the apparent strength of FRQ/CK1 interaction was affected?

STATISTICS/METHODS: The Methods say "All studies were performed on at least three independent experiments." Could the authors please confirm and clarify in the manuscript that the error bars are from multiple independent experiments (different days & different preparations)?

Reviewer #3 (Remarks to the Author):

Liu et al report a solid series of findings documenting an important role for the affinity of CK-1 for FRQ in determining period length.

Introduction: A relatively large amount of the text (Abstract 2nd sentence; Introduction: 2nd paragraph; 3rd paragraph 2nd & 3rd sentences; entire next-to-last paragraph; Discussion at several points), and a number of references (8-17, 23-26) are devoted to documenting the correlation between phosphorylation and protein stability even though, as the authors note, it has been known for several years that stability does not determine period. This emphasis misdirects the reader towards thinking in terms of turnover; the current view (2nd paragraph, end) could be emphasized earlier to better set the stage for the results, and the manuscript thereby shortened.

Figure 1a and b confirm prior knowledge; c and d show nicely that although FRQ-WC-2 interactions do not correlate with period length, FRQ-CK-1 interactions do so. 1e and f extends this correlation to classic frq alleles, providing an explanation for their period effects.

Figure 2 tests the logical hypothesis that if FRQ-CK-1 interactions are important for period determination, then mutations in regions previously shown to affect FRQ-CK-1 interactions should affect period length. They do, and interestingly some of the new mutations here show extreme period lengthening. The data also show that mutation of these helices do not affect the FRQ-WC interaction, and as anticipated that FRQ-CK1 interaction correlates with FRQ phosphorylation and stability (2e,f,g). Although, as the authors state, there is a roughly negative correlation between the FRQ-CK-1 interaction and period length, Figure 2d shows this is far from a linear correlation. If the error bars are really +/- 1 SD, then the mutants showing slightly long periods of about 30 hrs show interactions statistically similar to the one showing the extreme 48 hrs. It might be of interest to comment on this, or to leave out WT and expand the scale, including data from the intermediate strains such as E324D.

Figure 3 tests the important hypothesis that there is auto-feedback between CK-1 induced phosphorylation of FRQ and FRQ-CK-1 interaction. This is shown based on mutants (3a) and 3b confirms the corollary, that the daily cycle of FRQ phosphorylation must also entail a daily cycle in CK-1 interaction. In this context, the results in Figure S2 are somewhat surprising; the densitometry shows no cycle in FRQ-WC interaction whereas the gel suggests a cycle in binding and reference 9 also showed strong cycling. Figure 3cd extends the correlation to other mutants, and 3e nicely shows that phosphorylation of FRQ by other kinases (here CK2) can influence FRQ-CK-1 interactions with the attendant period changes; this is a good unifying observation. Figure 3f shows trypsin sensitivity data consistent with phosphorylation-induced structural changes in FRQ.

Figure 4 examines the effect of FRQ-CK-1 interaction on WC phosphorylation. Panels a and b show the two to be correlated, and c and d show them to be correlated to the amount of WC-2 binding to the frq promoter. IN c and d, the "CT" labels should be eliminated or a note made in the figure legend that they apply only to the WT strain and not to the mutants. Panel e shows that frq mutants with reduced FRQ-CK-1 interactions are not as effective at autoregulation. This ties up the story well, although the last paragraph repeats the already repeated finding that stability is not important.

Figure 5 presents a mathematical model consistent with the data. On page 11 the text says" Each

species is degraded with a rate of $[\delta]i$." I could not find this symbol in the equations; is this a typo?

Minor

The FLAG epitope is an octapeptide and should be capitalized throughout.

Response to Reviewers' comments:

Reviewer #1 (Remarks to the Author):

The authors present an intriguing hypothesis--that the stability of the CK1-FRQ complex in *Neurospora* (and potentially in other eukaryotic clocks) is a key determinant of period length. They provide a set of biochemical analyses and then a mathematical model in support of this hypothesis. Some of the results are quite compelling, including the generation of very long period alleles by rationally designing mutations in FRQ to disrupt interaction with CK1. However I feel that there are some serious weaknesses in the current version of this manuscript which make the results hard to interpret.

1. The hypothesis seems to predict (and this is supported by the authors' interpretation of their mathematical model) that instability of the CK1-FRQ complex will result in slowed accumulation of FRQ phosphorylation. When the complex is very weak, the period will lengthen dramatically because it is difficult to trip the positive feedback loop that enhances CK1-FRQ affinity. Thus a strong prediction is that the phosphorylation phase of the clock cycle will be specifically extended in mutants that decrease CK1-FRQ stability. This seems central to the study and should be tested directly.

Response: As suggested, we compared FRQ phosphorylation rhythms between the WT and one of long period FCD mutants (Q494N). As shown in Figure S2b, the FRQ phosphorylation phase in the Q494N mutant was markedly delayed compared to WT.

2. It is unclear what time points in the clock cycle were used for the interaction measurements e.g. in the IPs in Figure 1. Since the interaction between CK1 and FRQ is rhythmic, this should be clarified so we can be sure it is a fair comparison between different mutants which may show peak interaction at different times.

Response: Because FRQ phosphorylation is rhythmic in constant darkness (DD), we performed our IP experiments using cultures grown in constant light (LL) which has constant FRQ phosphorylation profile. This note is now added in main text and in the method.

3. The mathematical modeling suggests that changes in CK1 specific activity in the complex can also have dramatic effects on period. This is consistent with the idea that what really matters is the rate at which FRQ phosphorylation can accumulate. Is this inconsistent with the authors' conclusions? The text seems to focus on the parameters that determine complex stability.

Response: FRQ-CK1 complex formation precedes and is required for FRQ phosphorylation. Therefore, the stability of FRQ-CK1 complex determines the rate of FRQ phosphorylation. Thus, our data is consistent with the model prediction and elucidates a previously unappreciated role of the stability of FRQ-CK1 complex in determining circadian period.

minor points:

1. I'm not sure why the sensitivity analysis is limited to 3 parameter sets taken from the MCMC

distribution. Surely it would not be too computationally intensive to calculate a distribution of sensitivities for everything that fit well.

Response: The one- and two-parameter bifurcation analysis is a very labor-intensive process. We feel that randomly selected 3 parameter sets are sufficient. It should be noted that all 3 sets of parameters led to the same conclusion.

2. One potentially interesting prediction from this study is that mutants with decreased FRQ-CK1 stability may be more sensitive than wildtype to CK1 over expression (since this could drive the binding equilibrium towards complex formation)

Response: To address this concern, we inducibly over-expressed CK1a in the WT and FCD mutants (Q494N and V320I) to compare their period length changes due to an increase of CK1a level. Although the induction of CK1a resulted in period shortening in all strains, the FCD mutants were more sensitive to CK1a over expression than WT (Figure S2c). These results suggest that the binding equilibrium of the FRQ-CK1a complex formation is important for period determination.

Reviewer #2 (Remarks to the Author):

The manuscript “Formation of the FREQUENCY-CK1 complex underlies circadian period determination” by Liu et al. is a worthy addition to our understanding of the molecular daily clock in *Neurospora*, with potentially important ramifications to the mammalian and fly clock mechanisms. However, I have two concerns about the supporting evidence for the hypothesis presented in this manuscript:

1. The support for the hypothesis is basically the same kind of correlative support upon which the previous hypothesis (i.e., FRQ degradation rate) was based. In other words, previously the authors correlated period with FRQ degradation and claimed this correlation supported a causal link that FRQ degradation rate determined the circadian period. Now the authors say that this correlation breaks down with further examples, and their new hypothesis is that the FRQ/CK1 interaction is the cause. But the data are again correlative, not analytical. The authors set up the argument as though EITHER FRQ degradation rate is the cause OR the strength of FRQ/CK1 interaction is the cause. What about another possibility?

Response: The reviewer is correct that previous studies on the role of degradation on period were mostly correlative. Luis et al (reference 22) previously showed that in the *fwd-1* mutant, the period length and FRQ stability can be uncoupled, suggesting that FRQ phosphorylation but not FRQ degradation rate is the main determinant of period. In our study, we further confirmed this conclusion and showed that FRQ-CK1 interaction but not FRQ stability correlated in all major circadian period mutants we examined. Importantly, by creating a series of FRQ mutations in the FRQ-CK1 interaction domain, we showed that the FRQ-CK1 interaction plays a major role in period determination. In our FCD mutants, their period length correlated with their FRQ-CK1 interaction. In addition, we showed that FRQ phosphorylation acts by affecting FRQ-CK1a interaction. Finally, our conclusion is confirmed by mathematic modeling study that includes all

major known processes of the circadian feedback loop. Therefore, our conclusion is based on much more than just correlative data.

It should be noted that other processes in the circadian feedback loops, such as WC levels and WC phosphorylation independent of FRQ, can also contribute to period determination. We have now added this statement in the discussion.

These correlations are largely based on mutants that are already known to have period defects, and there is only a little work with mutants that were first identified to alter FRQ/CK1 interaction to “hypothesis test” their ideas. (e.g., mutations in the FCD domains caused long period, but unfortunately there are no mutations in either FCD1 or FCD2 that can enhance the FRQ-CK1a interaction and cause shorter period phenotypes, which would be a persuasive demonstration.) Moreover, are there any mutants that affect FRQ/CK1 interaction that do NOT cause period defects? Frankly, I think their presentation would be enhanced if they reorganize to emphasize hypothesis testing with previously unreported mutations in the FRQ/CK1 interaction domains to test whether those interaction domain mutants also exhibit period effects. This would be a more persuasive approach to present the hypothesis and data.

Response: Vast majority of mutations in a protein complex will only compromise the interaction. We consulted with several structure experts on the possibility of predicting potential mutations that can strengthen the FRQ-CK1 interaction. However, based on current structure prediction methods, it is nearly impossible to predict mutations that can enhance interaction without the availability of high-resolution structure of the protein complex. Enhanced interaction may need mutations of the interacting amino acids on both FRQ and CK1a. In the future, we will try obtain high-resolution structure of the FCD-CK1.

For our FCD mutants, the period length correlated with the FRQ-CK1 interaction. We did not find any mutants that affect CK1-FRQ interaction but do not have period changes.

As suggested, we have re-organized the introduction of the paper to highlight the potential importance of the CK1a interaction with clock proteins.

2. What does “interaction” mean to these authors? Isn’t “affinity” and/or “binding constant” the relevant parameter? Is the relative amount of immunoprecipitated protein really the same thing as “strength” of interaction? The chronobiology field is moving to more and more rigorous biochemical criteria, and the amount of immunoprecipitated protein just doesn’t seem to be a rigorous demonstration of affinity. How do they calculate and normalize the interaction? The Fig. 1 legend states “Quantification of relative FRQ-WC-2 interaction levels is based on the ratio of IP to Input.” How do the authors correct for loading errors and sample-to-sample variability of IP efficiency?

Response: The affinity of protein-protein interaction determines the amount of immunoprecipitated protein. Since our method did not directly measure “affinity”, we used “strength” to describe the interaction. For the quantification method of the IP assay, the relative FRQ-CK1a and FRQ-WC-2 interaction levels are based on the ratio of immunoprecipiated FRQ

to Input FRQ, and normalized by CK1a and WC-2 levels, respectively.” We clarified this in the revised methods.

Can the authors purify these proteins and form a complex in vitro? How about testing interaction of native vs. mutant proteins where one protein is immobilized on a Biacore surface and the interaction is tested by SPR? Perhaps CK1 could be immobilized and the IP-complex flowed over the surface and an affinity SPR measurement performed? (I recognize that the complex might not bind to CK1 on the surface because CK1 in the complex is masking the interaction sites, but perhaps the authors can think of a better way to accomplish the same goal.)

Response: This is a great idea but a very difficult experiment to do. It should be noted that most of FRQ protein is predicted to be unstructured. Our lab previously tried very hard for several years to purify the recombinant FRQ-FRH complex. However, the complex quickly aggregates and become impossible to carry out further biochemical assays that are biologically meaningful.

Minor Points/Suggestions:

Fig. 2d: The correlation between “interaction” (ordinate) and period (abscissa) is rather catastrophic (exponential); there aren’t enough intermediate values of period to make this correlation persuasive.

Response: We have performed IP assays using additional strains with different period lengths. The result is now included in the revised Figure 2d.

Fig. 3b (right): why does the period seem to be so short?

Response: The period of *Neurospora* is about 22h. These experiments had samples were only collected every 6 hours so it is difficult to estimate the period. The period phenotype is much clear in our rhythmic experiments in Figure S2b.

Fig. 3c/d: only a few mutants are shown here. Don’t you have more mutants available to test these correlations? The differences are small and/or variable and not very persuasive.

Response: We tested more long period mutants than shown here and the results are also consistent with our conclusion. Here we only showed results in which both long and short period mutants were compared side-by-side with the wild-type. The sensitivity of the IP assay and somewhat modest period changes in these mutants limited the difference we could observe. Although these differences are modest, they are significant.

Fig. 4a/b vs. 4c/d: the effects of the mutations on phosphorylation (a/b) are small and subtle, whereas the ChIP effects (c/d) appear to be quite large. It does not appear that these two effects are directly correlated.

Response: The mobility shift difference just indicate phosphorylation changes and is not an accurate method to examine WC binding activity. So, ChIP assay is really a good method for measuring WC binding activity.

Fig. 5: why has modeling only just now revealed that the rate of FRQ degradation minimally affects period? What is different about this model and why did previous models indicate that FRQ degradation could/would have a significant effect on period? Other than the current study that precipitated this new model, are there any particularly designed experiments that support the proposed mathematical modeling and computational simulations for the FRQ-CK1a interaction and period determination?

Response: Our mathematic model is based on previous reported model (references 59&60). Briefly, our model was able to identify FRQ-CK1 complex as the key period determining mechanism because of newly introduced positive feedback loop where FRQ-CK1 complex promoting the association of FRQ and CK1. This is first proposed by us and have not reported by others. We have communicated with other groups and got positive feedback from them, because CK1-FRQ/PER interaction could also explain their period mutants.

DISCUSSION: The authors should address in the Discussion possible alternative interpretations of their data. Also, did the authors test any of the fwd-1 mutants (from Larrondo et al. 2015) to determine if the apparent strength of FRQ/CK1 interaction was affected?

Response: We revised our Discussion and add additional processes can also contribute to period determination. We previously reported (reference 54) that FRQ-CK1 interaction was indeed affected in the fwd-1 mutant.

STATISTICS/METHODS: The Methods say “All studies were performed on at least three independent experiments.” Could the authors please confirm and clarify in the manuscript that the error bars are from multiple independent experiments (different days & different preparations)?

Response: Yes. Clarified in the methods as suggested.

Reviewer #3 (Remarks to the Author):

Liu et al report a solid series of findings documenting an important role for the affinity of CK-1 for FRQ in determining period length.

Introduction: A relatively large amount of the text (Abstract 2nd sentence; Introduction: 2nd paragraph; 3rd paragraph 2nd & 3rd sentences; entire next-to-last paragraph; Discussion at several points), and a number of references (8-17, 23-26) are devoted to documenting the correlation between phosphorylation and protein stability even though, as the authors note, it has been known for several years that stability does not determine period. This emphasis misdirects the reader towards thinking in terms of turnover; the current view (2nd paragraph, end) could emphasized earlier to better set the stage for the results, and the manuscript thereby shortened.

Response: The reviewer is correct that a previous study in *Neurospora* suggested that FRQ phosphorylation but not protein stability plays a more important role in period determination. However, the role of clock protein turnover in other clock systems was not clear. Because our study here has important implications in animal circadian systems, we feel that some background

on the topic will be helpful for readers that are not very familiar with the *Neurospora* literatures. We have now re-organized and shortened the introduction of the paper.

Figure 1a and b confirm prior knowledge; c and d show nicely that although FRQ-WC-2 interactions do not correlate with period length, FRQ-CK-1 interactions do so. 1e and f extends this correlation to classic frq alleles, providing an explanation for their period effects.

Figure 2 tests the logical hypothesis that if FRQ-CK-1 interactions are important for period determination, then mutations in regions previously shown to affect FRQ-CK-1 interactions should affect period length. They do, and interestingly some of the new mutations here show extreme period lengthening. The data also show that mutation of these helices do not affect the FRQ-WC interaction, and as anticipated that FRQ-CK1 interaction correlates with FRQ phosphorylation and stability (2e,f,g). Although, as the authors state, there is a roughly negative correlation between the FRQ-CK-1 interaction and period length, Figure 2d shows this is far from a linear correlation. If the error bars are really ± 1 SD, then the mutants showing slightly long periods of about 30 hrs show interactions statistically similar to the one showing the extreme 48 hrs. It might be of interest to comment on this, or to leave out WT and expand the scale, including data from the intermediate strains such as E324D.

Response: As suggested, we performed IP using additional mutants. The results are now shown in the revised Figure 2d.

Figure 3 tests the important hypothesis that there is auto-feedback between CK-1 induced phosphorylation of FRQ and FRQ-CK-1 interaction. This is shown based on mutants (3a) and 3b confirms the corollary, that the daily cycle of FRQ phosphorylation must also entail a daily cycle in CK-1 interaction. In this context, the results in Figure S2 are somewhat surprising; the densitometry shows no cycle in FRQ-WC interaction whereas the gel suggests a cycle in binding and reference 9 also showed strong cycling. Figure 3cd extends the correlation to other mutants, and 3e nicely shows that phosphorylation of FRQ by other kinases (here CK2) can influence FRQ-CK-1 interactions with the attendant period changes; this is a good unifying observation. Figure 3F shows trypsin sensitivity data consistent with phosphorylation-induced structural changes in FRQ.

Response: Quantification of relative FRQ-WC-2 interaction levels in the original Figure S2 (revised Figure S3) is based on the ratio of the level of immunoprecipitated FRQ to the Input FRQ and was normalized by WC-2 levels. Similar results were observed in several independent experiments. It is likely that a difference in sensitivity between our IP method and the previously performed MS method might cause the different results between our study and that in reference 9.

Figure 4 examines the effect of FRQ-CK-1 interaction on WC phosphorylation. Panels a and b show the two to be correlated, and c and d show them to be correlated to the amount of WC-2 binding to the frq promoter. IN c and d, the “CT” labels should be eliminated or a note made in the figure legend that they apply only to the WT strain and not to the mutants. Panel e shows that frq mutants with reduced FRQ-CK-1 interactions are not as effective at autoregulation. This ties

up the story well, although the last paragraph repeats the already repeated finding that stability is not important.

Response: We revised the labels in Figure 4 as suggested.

Figure 5 presents a mathematical model consistent with the data. On page 11 the text says "Each species is degraded with a rate of $[\delta]_i$." I could not find this symbol in the equations; is this a typo?

Response: The i in δ_i indicates 1-10. We now clarified this in the text.

Minor

The FLAG epitope is an octapeptide and should be capitalized throughout.

Response: Revised as suggested.

Reviewers' comments:

Reviewer #1 (Remarks to the Author):

I think that this is a valuable study that makes an important contribution to the study of circadian mechanism and should be published. The experimental additions to the paper in revision strengthen the work substantially. However, I am bit dismayed by the approach of the authors towards the modeling part of the study which suggests that mathematical modeling is really only being used here to the extent that it can confirm the original hypothesis.

I said: "The mathematical modeling suggests that changes in CK1 specific activity in the complex can also have dramatic effects on period. This is consistent with the idea that what really matters is the rate at which FRQ phosphorylation can accumulate. Is this inconsistent with the authors' conclusions? The text seems to focus on the parameters the determine complex stability."

Response: FRQ-CK1 complex formation precedes and is required for FRQ phosphorylation. Therefore, the stability of FRQ-CK1 complex determines the rate of FRQ phosphorylation. Thus, our data is consistent with the model prediction and elucidates a previously unappreciated role of the stability of FRQ-CK1 complex in determining circadian period.

o.k., but in the table in Figure 5b, it clearly shows that the model predicts that the specific activity in the complex (α_2) will also have a dramatic effect on period. The authors do include a paragraph in the revised text indicating that there are many influences on period, but I still feel like the message the model is hinting at is that the total rate of change of FRQ phosphorylation is what's important (rather than binding affinity only, as the title of the paper suggests)

I said: I'm not sure why the sensitivity analysis is limited to 3 parameter sets taken from the MCMC distribution. Surely it would not be too computationally intensive to calculate a distribution of sensitivities for everything that fit well.

Response: The one- and two-parameter bifurcation analysis is a very labor-intensive process. We feel that randomly selected 3 parameter sets are sufficient. It should be noted that all 3 sets of parameters led to the same conclusion.

I'm not sure why the authors say that all 3 sets of parameters lead to the same conclusion: in one set, α_2 is the most sensitive and in the others α_1 is the most sensitive. Given the amount of labor that went into the experimental part of this study, it seems reasonable to provide a thorough statistical analysis of the model sensitivity--e.g. is there something systematically different about parameter sets where α_2 is more sensitive.

Reviewer #2 (Remarks to the Author):

The revised manuscript "Formation of the FREQUENCY-CK1 complex underlies circadian period determination" by Liu et al. is a significant improvement over the originally submitted manuscript that adequately addresses most of this reviewer's concerns.

Reviewer #3 (Remarks to the Author):

As noted by reviewer #2, statements made in the text often seem to confuse correlation with proof in the interest of emphasizing the importance of particular aspects of the data; still, the underlying data is broad of interest. However, these statements can get the authors into trouble. For instance, the Discussion states (incorrectly) that " hyperphosphorylated FRQ cannot efficiently form a complex with CK-1a" (Figure 3 shows that the reduction is more on the order of 3 fold), but this raises an interesting point:

If (1) phosphorylation of FRQ affects complex formation with CK-1a, and (2) CK-1a only stably associates with WCC via FRQ, then how does (3) CK-1a not have a rhythm in association with WC-2 as Figure S3a shows? They cannot all be correct, and if so, which of those statements, 1, 2, or 3, is wrong? Whichever one is wrong ought to be corrected in the relevant text and figures.

Response to Reviewers' comments

Reviewer #1 (Remarks to the Author):

I think that this is a valuable study that makes an important contribution to the study of circadian mechanism and should be published. The experimental additions to the paper in revision strengthen the work substantially. However, I am bit dismayed by the approach of the authors towards the modeling part of the study which suggests that mathematical modeling is really only being used here to the extent that it can confirm the original hypothesis.

I said: "The mathematical modeling suggests that changes in CK1 specific activity in the complex can also have dramatic effects on period. This is consistent with the idea that what really matters is the rate at which FRQ phosphorylation can accumulate. Is this inconsistent with the authors' conclusions? The text seems to focus on the parameters that determine complex stability."

Response: FRQ-CK1 complex formation precedes and is required for FRQ phosphorylation. Therefore, the stability of FRQ-CK1 complex determines the rate of FRQ phosphorylation. Thus, our data is consistent with the model prediction and elucidates a previously unappreciated role of the stability of FRQ-CK1 complex in determining circadian period.

o.k., but in the table in Figure 5b, it clearly shows that the model predicts that the specific activity in the complex (α_2) will also have a dramatic effect on period. The authors do include a paragraph in the revised text indicating that there are many influences on period, but I still feel like the message the model is hinting at is that the total rate of change of FRQ phosphorylation is what's important (rather than binding affinity only, as the title of the paper suggests)

Response: There are two rate constants that describe rate of phosphorylation of FRQ in our model: the rate of initial phosphorylation of FRQ by CK1, α_2 , and the rate of additional phosphorylation of FRQ in its complex form with CK1 and WCC, α_4 , where we assume subsequent phosphorylation and inactivation of WCC. In all 3 parameter sets, we show that α_4 is not a sensitive parameter and does not change the period of circadian rhythms (Figure S5), which indicates that the impact of the rate of initial phosphorylation of FRQ, α_2 , is distinct from α_4 . The critical difference between α_2 and α_4 is that α_2 is one of the key parameters that determine the nonlinear dynamics of the model due to the positive feedback loop (i.e. FRQp-CKI promoting the association of FRQ-CK1 complex), and dramatically changes the circadian period when it's perturbed.

However, we acknowledge that our model is a relatively simple model, which does not account for extensive phosphorylation of FRQ over a circadian cycle. Therefore, we agree with the reviewer that 'the total rate of change of FRQ phosphorylation' may be critical in determining the period of circadian rhythms, and we changed the title of our manuscript to "Formation of the FREQUENCY-CK1 complex is a major process that determines the period of circadian rhythms".

I said: I'm not sure why the sensitivity analysis is limited to 3 parameter sets taken from the

MCMC distribution. Surely it would not be too computationally intensive to calculate a distribution of sensitivities for everything that fit well.

Response: The one- and two-parameter bifurcation analysis is a very labor-intensive process. We feel that randomly selected 3 parameter sets are sufficient. It should be noted that all 3 sets of parameters led to the same conclusion.

I'm not sure why the authors say that all 3 sets of parameters lead to the same conclusion: in one set, alpha_2 is the most sensitive and in the others alpha_1 is the most sensitive. Given the amount of labor that went into the experimental part of this study, it seems reasonable to provide a thorough statistical analysis of the model sensitivity--e.g. is there something systematically different about parameter sets where alpha_2 is more sensitive.

Response: We apologize if our explanation was not clear regarding our conclusions. Our results showed that all 3 randomly selected sets of default parameters lead to the same conclusion, because all of them identified alpha_c, alpha_1, and alpha_2 as the top 3 sensitive parameters, which are critical parameters determining the dynamics of the positive feedback loop in the model.

There might be a confusion between statistical analysis to search a large number of default parameter sets using MCMC and *period sensitivity analysis* for the 3 parameter sets using bifurcation analysis. Below, we clarify distinct usages of MCMC approach and bifurcation analysis for different objectives.

First, we ran MCMC chain for 4,000 iterations to obtain multiple sets of default parameters that reproduce cyclic bioluminescent data reflecting the activity of *frq* promoter. Distributions of parameters are summarized in the newly added supplemental figure 5. We observe that each parameter has different spread with distinct mean values. For example, alpha_5 shows a large spread of parameter space ranging from close to 0 to ~11 with a mean value close to 0. This indicates that alpha_5 is not a critical parameter in reproducing experimental data, because oscillations can be generated even when alpha_5 is close to 0 (with a wide range of values of alpha_5). In contrast, delta_1 and delta_5 show tighter spread of parameter space, which suggest that they may play critical roles in generating autonomous oscillations. However, distributions of these parameters do not indicate how period is altered as a function of each parameter in the system.

The objective of bifurcation analysis is to measure how the period changes as a function of each parameter. For univariate bifurcation analysis, the value of a single parameter (k_i) is changed to compute corresponding period while other parameters are fixed as default values (Figure S5). This process (i.e. bifurcation analysis) is labor intensive, and it is technically not feasible to provide a distribution of parameters from a large number of parameter sets that reproduce experimental data. Furthermore, to the best of our knowledge, no statistical analysis has been developed to summarize the robustness of the bifurcation analysis (i.e. period sensitivity analysis) against the changes in default parameters. The main difficulty in developing a statistical measure is from the fact that the bifurcation analysis derives its conclusion directly from graphs, rather than using a single measure. Developing a new method to check the robustness of bifurcation

analysis using statistical tools will require significant mathematical advancements integrating bifurcation and statistical analyses, which is beyond the scope of this paper, but we view this as a very interesting future research topic.

To better clarify our explanation, we have now added the result described above as Supplemental Figure 5 and revised the text and method section on our mathematic modeling and *in silico* simulation experiments.

Reviewer #2 (Remarks to the Author):

The revised manuscript “Formation of the FREQUENCY-CK1 complex underlies circadian period determination” by Liu et al. is a significant improvement over the originally submitted manuscript that adequately addresses most of this reviewer’s concerns.

Response: We appreciate the support from this reviewer.

Reviewer #3 (Remarks to the Author):

As noted by reviewer #2, statements made in the text often seem to confuse correlation with proof in the interest of emphasizing the importance of particular aspects of the data; still, the underlying data is broad of interest. However, these statements can get the authors into trouble. For instance, the Discussion states (incorrectly) that “ hyperphosphorylated FRQ cannot efficiently form a complex with CK-1a” (Figure 3 shows that the reduction is more on the order of 3 fold), but this raises an interesting point:

If (1) phosphorylation of FRQ affects complex formation with CK-1a, and (2) CK-1a only stably associates with WCC via FRQ, then how does (3) CK-1a not have a rhythm in association with WC-2 as Figure S3a shows? They cannot all be correct, and if so, which of those statements, 1, 2, or 3, is wrong? Whichever one is wrong ought to be corrected in the relevant text and figures.

Response: The reviewer might have misunderstood the result of Figure S3a. In the experiment described in Figure S3a, we examined that the interaction between FRQ and WC-2 and did not examine the interaction between CK1a and WC-2. Our result showed that the amount of FRQ was rhythmic but the FRQ-WC-2 interaction was not significantly affected by FRQ phosphorylation level. The results from our lab and other labs showed that FRQ acts as the scaffold between CK1a and WCC and rhythmic binding between FRQ and CK1a results in rhythmic WC phosphorylation. We have now clarified this in the revised manuscript.

REVIEWERS' COMMENTS:

Reviewer #1 (Remarks to the Author):

The authors have addressed my concerns with the inclusion of the new supplemental fig 5. I think the paper should be published in its current form.

point-by-point response to reviewer's comment

Reviewer #1:

The authors have addressed my concerns with the inclusion of the new supplemental fig 5. I think the paper should be published in its current form.

Response: We are pleased that the reviewer is now satisfied with our revision.